# Towards Federated Foundation Models: Scalable Dataset Pipelines for Group-Structured Learning

**Zachary Charles**[*]
Google Research
zachcharles@google.com

**Nicole Mitchell**[*]
Google Research
nicolemitchell@google.com

**Krishna Pillutla**[*]
Google Research
kpillutla@google.com

**Michael Reneer**
Google Research
michaelreneer@google.com

**Zachary Garrett**
Google Research
zachgarrett@google.com

## Abstract

We introduce Dataset Grouper, a library to create large-scale group-structured (e.g., federated) datasets, enabling federated learning simulation at the scale of foundation models. This library facilitates the creation of group-structured versions of existing datasets based on user-specified partitions, and directly leads to a variety of useful heterogeneous datasets that can be plugged into existing software frameworks. Dataset Grouper offers three key advantages. First, it scales to settings where even a single group's dataset is too large to fit in memory. Second, it provides flexibility, both in choosing the base (non-partitioned) dataset and in defining partitions. Finally, it is framework-agnostic. We empirically demonstrate that Dataset Grouper enables large-scale federated language modeling simulations on datasets that are orders of magnitude larger than in previous work, allowing for federated training of language models with hundreds of millions, and even billions, of parameters. Our experimental results show that algorithms like FedAvg operate more as meta-learning methods than as empirical risk minimization methods at this scale, suggesting their utility in downstream personalization and task-specific adaptation. Dataset Grouper is available at https://github.com/google-research/dataset_grouper.

## 1 Introduction

In most machine learning and artificial intelligence settings, algorithms operate on "flat" collections of examples, that is, examples with no differentiation in source or group structure. However, data in the real world often consists of an explicit or implicit underlying *group structure*, where the examples are partitioned across some number of groups with markedly different statistical characteristics. Increasingly, research has found that this structure is important in a variety of settings, both for encoding constraints (such as data restrictions) and in developing algorithms for learning tasks.

Federated learning (FL) is one such setting. FL methods are designed to operate on data partitioned explicitly across "clients". In cross-device FL [1, Table 1], clients are often edge devices which exhibit heterogeneity in both quantity and distribution of data. Cross-silo FL exhibits a similar structure, often with a coarser notion of clients (such as institutions or companies). Group structures also arise in meta-learning, in which data is grouped according to a notion of "task" [2], and in personalization, in which a user's specific data is used to tune an algorithm's outputs. The group structure can also be highly relevant in the context of differential privacy [3, 4]. An intuitive "unit

---

[*]Authors contributed equally to this work.

37th Conference on Neural Information Processing Systems (NeurIPS 2023) Track on Datasets and Benchmarks.

| Source | Dataset | Group by | Words | Groups | Words per group | | | Examples | Words per example | | |
|---|---|---|---|---|---|---|---|---|---|---|---|
| | | | | | 10th perc. | Median | 90th perc. | | 10th perc. | Median | 90th perc. |
| **Ours** | FedC4 | Domain | **132B** | **15.6M** | 82 | 815 | **0.5B** | **0.36B** | 49 | 191 | 783 |
| | FedWiki | Article | 3B | 6.5M | 39 | 198 | 70K | 6.5M | 39 | 198 | 70K |
| | FedBookCO | Book | 1.2B | 18K | **24K** | **52K** | 4M | 18K | **24K** | **52K** | **4M** |
| | FedCCnews | Domain | 0.3B | 8.8K | 303 | 5K | 8.4M | 0.7M | 78 | 316 | 842 |
| **Existing** | Amazon Reviews | Account | 4.3B | 1.5M | 278 | 1.1K | 5K | 68M | 3 | 28 | 155 |
| | Stack Overflow | Account | 2B | 0.3M | 1.2K | 2.7K | 11K | 0.1B | 3 | 13 | 29 |
| | Reddit | Account | 1.2B | 1.7M | 58 | 257 | 1720 | 33M | 7 | 21 | 81 |
| | Blog Corpus | Account | 0.1B | 17K | 551 | 2K | 13K | 0.5M | 6 | 105 | 460 |
| | Shakespeare | Role/play | 0.4M | 715 | 14 | 175 | 1.6K | 16K | 4 | 12 | 63 |
| | Gigaword | Synthetic | 0.3M | 100 | 3.0K | 3.1K | 3.2K | 10K | 21 | 31 | 41 |

Table 1: Summary of the per-group (i.e., per-client) and per-example (i.e., per-sequence) statistics of the new language modeling datasets we introduce using Dataset Grouper, compared to those of previous federated benchmark datasets supplied by TFF [11], LEAF [12], FedNLP [13, 14], and FedScale [15].

of privacy" is the total collection of examples associated with a given user [5]. To ensure user-level differential privacy, we must generally train the model in a user-aware manner.

The increasing prominence of foundation models and large language models (LLMs) and their wise applicability to downstream tasks enhance the need for group-structured data. Though foundation models are generally trained on massive flat datasets, they are often evaluated by considering the performance on various benchmarks, yielding a natural group structure. Moreover, for downstream, user-facing applications, one may want to train on user-partitioned data that is representative of the actual task at hand. Alternatively (or in conjunction) one can personalize a foundation model for a given user [6–8]. In all these settings, one may wish to maintain formal user-level privacy guarantees, especially given the privacy and memorization concerns surrounding foundation models [9, 10].

All of the aforementioned scenarios require datasets with explicit group structure. Since foundation models generally require large amounts of data, these research areas may specifically benefit from large-scale group-structured datasets. Unfortunately, to the best of our knowledge, there are relatively few existing datasets that meet such criteria. While a variety of federated datasets are available to researchers [11–15], many of these are small-scale in terms of the number of groups, the quantity of data, or quantity of data per group. Moreover, they may only be available in formats that do not scale well, either due to memory requirements or insufficient efficiency.

**Contributions.** In this work, we address the growing need for a wider variety of group-structured datasets, especially at larger scales. Concretely, we make the following contributions.

- **A library for creating group-structured datasets**: We introduce Dataset Grouper, a library that can *flexibly* create group-structured (and federated) versions of existing datasets via user-defined partition functions. We engineer it for *efficiency*, both in partitioning datasets and in iterating over data. The library is designed with large-scale datasets in mind and can support datasets with millions or even billions of groups. Dataset Grouper can be used to create group-structured versions of all datasets available in Tensorflow Datasets [16] and HuggingFace Datasets [17].

- **Large-scale federated text datasets**: While Dataset Grouper can be used for a wide array of modalities and tasks, we illustrate its use by creating group-structured versions of four large language modeling datasets (see Table 1 and Figure 1), designed specifically for FL research. They are orders of magnitude larger than previous datasets in terms of one or more of the following: the number of groups, the amount of data, and the length of sequences. They are suitable for both pre-training and fine-tuning tasks, and exhibit long tails, as is common in large-scale text corpora.

- **Experiments**: We train $O(100M)$ and $O(1B)$ parameter decoder-only transformer models from scratch on a group-structured version of the C4 dataset [18], using FL training algorithms. This is, to the best of our knowledge, the first demonstration of federated training of a model of this magnitude on a federated dataset of this scale. We compare FedAvg and FedSGD in this setting in terms of their pre- and post-personalization metrics. Our results highlight that at this scale, FedAvg behaves more like a meta-learning algorithm than an empirical risk minimization algorithm.

**On the term "federated".** Our work is primarily motivated by the research needs of the FL community. Throughout, we will often approach questions, design decisions, and experiments from this

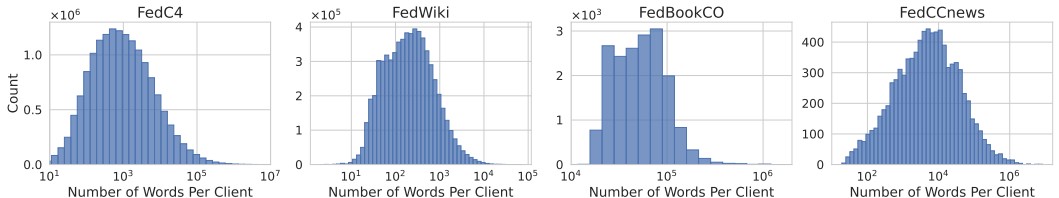

Figure 1: Per-group statistics of the new group-structured (i.e. federated) language modeling datasets.

perspective, and will occasionally use the terms "federated" and "group-structured" interchangeably. This is a broadening of the definition of the federated setting often given in the literature, especially the definition proposed by Kairouz et al. [19]. There and in previous literature, FL is characterized by the group-level structure of the data coupled with the location of each group (e.g., a client's dataset is assumed to reside only on its local device). In this work, we de-emphasize the location of the data, and will primarily focus on the group-structure of the data regardless of where that data lives.

## 2    Related Work

Many widely used group-structured datasets arise from the FL community. Early work on FL benchmark datasets combined benchmark datasets with simulation environments. For example, LEAF [12], TensorFlow Federated (TFF) [11], Flower [20], FedML [14] (and its text-specific dataset provided in FedNLP [13]), FedScale [15], FLBench [21], OARF [22], and FedJAX [23] all supply partitioned datasets commonly used in FL benchmarks. These frameworks have helped drive much of FL research into areas such as optimization algorithms [24–31], privacy and security [32–35], robustness to adversaries [36–40] and distribution shifts [41–45], and personalization [46–51].

Later work on FL introduced specialized datasets and benchmarks. FLAIR [52] is a large-scale multi-label image classification dataset with $0.4M$ images from 50K Flickr accounts. FLamby [53] is a cross-silo FL benchmark for healthcare applications where the datasets contain 2-6 groups with 400 to 23K total examples. The personalized federated learning algorithm benchmark Motley [54]'s largest dataset is Stack Overflow while pFL-bench [55] offers no language modeling datasets.

The scale of the existing language modeling datasets that FL frameworks provide is summarized in Table 1. By comparison, the datasets we provide are significantly larger, which allows training models that are an order of magnitude larger than previous work [56]. Additionally, our datasets are generally framework-agnostic and can be combined with many of these simulation frameworks. Further, existing federated datasets that are derived from datasets in TensorFlow Datasets [16], such as Amazon Reviews and Blog Corpus, can be generated via Dataset Grouper.

Group-structured data have also been studied in the context of distribution shifts, e.g., the WILDS benchmark [57, 58]. The datasets provided in WILDS are smaller than what we consider in Table 1 — the code language modeling dataset Py150 dataset has 150K examples partitioned over 8K groups. Moreover, WILDS does not support optimized per-group data-loaders as of this writing [59], which are necessary for benchmarking federated algorithms.

LLMs are typically pre-trained on large web-scraped text corpora without any group structure [e.g. 18, 60, 61]. Besides the tremendous amount of data on which they are trained [62], the success of LLMs is also driven by the capacity of these models to handle much longer sequences than previous RNN-based models [63–65]. This requires datasets with long enough contiguous sequences that contain hundreds to thousands of words. Almost all of the existing group-structured language modeling datasets have extremely short sequences (Table 1). For instance, the Stack Overflow dataset has a median and 90th percentile sequence lengths of 13 and 29 words respectively. In comparison, the datasets we introduce have significantly longer sequences, e.g., FedBookCO has on the order of $10^3$ to $10^6$ words per sequence.

Some recent advances at the intersection of FL and foundation models include collaborative prompt tuning using FL [66], federating chain-of-though reasoning [67] through diverse crowd-workers [68], and instruction-tuning LLMs using FL [69]. Dataset Grouper can also be used to generate federated datasets compatible with these methods as well.

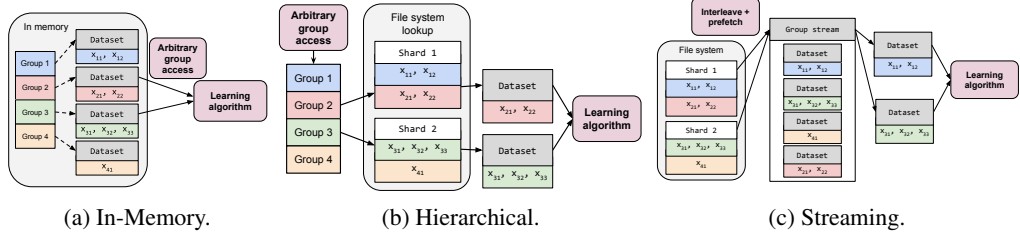

| (a) In-Memory. | (b) Hierarchical. | (c) Streaming. |

Figure 2: High-level representations of group-structured dataset formats.

# 3  Core Design

We now discuss the core design of Dataset Grouper and the various trade-offs it entails. One unifying theme throughout is a focus on enabling the types of training workflows used for foundation models, at the expense of some amount of flexibility in what kinds of simulations can be performed.

## 3.1  Group-Structured Datasets at Scale

Our primary goal is to enable research using large-scale group-structured datasets. In order to do so, we need a group-structured dataset format that balances the following characteristics:

- **Scalability**: Can the dataset format scale to large numbers of examples, groups, and bytes?
- **Group access time**: How long does it take to access the examples held by a single group?
- **Group access patterns**: What kinds of sampling patterns across groups are permitted? Can we access group datasets arbitrarily, and in any order?

There is a trade-off between these characteristics. Dataset formats used in the FL community often optimize for either scalability or group sampling time, while enabling maximum flexibility in access patterns. Our core insight is that by limiting the access patterns possible, we can use a dataset format that is scalable and efficient simultaneously. We discuss three archetypes of group-structured dataset formats (in-memory, hierarchical, and streaming) and their resulting trade-offs briefly in Table 2, and in more detail below. Figure 2 gives a graphical representation of the formats.

**In-memory formats.** In-memory group-structured datasets are essentially key-value mappings held entirely in memory. Adopted by e.g. LEAF [12] and FedNLP [13], this is suitable for small datasets such as EMNIST or CIFAR-100. Looking up a group's dataset is fast (e.g., via a hash map), and groups can be accessed in an arbitrary order. Of course, this approach is limited by the cumulative size of the dataset and is therefore not scalable in full generality. As we see in Table 3, this format does not even scale to FedBookCO on a single CPU; FedC4 and FedWiki are even larger.

**Hierarchical formats.** Hierarchical dataset formats store examples across files in such a way that (a) the dataset need not be loaded entirely into memory, and (b) individual groups can be constructed in arbitrary orders. For example, TensorFlow Federated [11] uses SQL databases to both store and access client datasets for FL simulations, facilitating the loading of the group index in-memory, then construction of a group's dataset at a later time. For larger datasets, constructing an arbitrary group's dataset can be slow, as it is often bottlenecked by indexing and searching over a large number of (possibly distributed) files. Table 3 shows that the hierarchical format can be significantly slower than other formats when accessing groups in very large datasets.

|  | **In-Memory** | **Hierarchical** | **Streaming** |
|---|:---:|:---:|:---:|
| Scalability | Limited | High | High |
| Group Access Time | Very Fast | Slow | Fast |
| Group Access Patterns | Arbitrary | Arbitrary | Shuffle + Streaming |

Table 2: Characteristics of group-structured dataset formats.

| Dataset Format | In-Memory | Hierarchical | Streaming |
|---|---|---|---|
| CIFAR-100 | $0.0783 \pm 0.0007$ | $25.11 \pm 0.81$ | $9.88 \pm 0.075$ |
| FedCCnews | $0.549 \pm 0.014$ | $> 7200$ | $248 \pm 17.5$ |
| FedBookCO | Out of memory | $> 7200$ | $192 \pm 9.07$ |

Table 3: The time (in seconds) to iterate over federated datasets. This is the time required to iterate over all examples in all group datasets, in serial, on a single CPU. We present the average and standard deviation over 5 trials, omitting trials that take more than 2 hours ($> 7200$ seconds), or that ran out of memory. We compare a federated CIFAR-100 dataset (partitioned across 100 groups, each with 100 examples), FedCCnews (in which examples are split across users at a domain level), and FedBookCO (in which examples are split across users at a title level). See Section 4 for more details on the latter two datasets.

**Streaming formats.** Instead of allowing arbitrary group access, Dataset Grouper provides ways to iterate over all the groups in a stream-like fashion. The datasets for each group are backed by some number of files,[2] which are interleaved to create a "group stream". Concretely, this restricts the possible group access patterns, only allowing stream-level operations such as buffered shuffling, repeating, and batching. This essentially lifts the stream-of-examples format used large-scale centralized training pipelines to streams of groups for federated training — both formats allow dataset iterators with limited shuffling (e.g., with a fixed-size buffer), but not arbitrary access to the individual elements. This restriction allows us to use parallel reads, prefetching, and interleaving to speed up dataset iteration and generally enables the total iteration time of the dataset to scale linearly (as opposed to super-linearly) with the number of groups in the dataset.

Each group's dataset is further represented as a stream of examples so that no group's data need to be fully loaded into memory. This is crucial in scaling to large datasets like FedC4, something that is memory-prohibitive for in-memory formats, and speed-prohibitive for hierarchical formats. To illustrate this further, we detail the time required to iterate fully over various group-structured datasets (accessing the groups' datasets sequentially, in a random order) in different formats in Table 3. For details on the amount of memory used by each format, see Appendix E.

## 3.2 Flexible and Efficient Dataset Partitioning

An underlying theme of both foundation model research and FL research is the need for a wide variety of datasets. It is often useful to have different datasets for different downstream tasks and modalities for foundation models, while the wide variety of FL settings (e.g. cross-device vs. cross-silo) and types of group heterogeneity (feature heterogeneity, label heterogeneity, heteroskedasticity, etc.) require dedicated datasets. It is often useful in FL to be able to explicitly partition the same dataset in multiple ways, in order to understand the impact of heterogeneity [71]. Therefore, our second key design goal is to allow flexibility both in the base (non-partitioned) dataset and in how it is partitioned.

To achieve this, we make two important, albeit related, design decisions. First, Dataset Grouper does not directly host datasets, but instead allows users to create partitioned versions of existing datasets in TensorFlow Datasets [16] and HuggingFace Datasets [17]. Second, Dataset Grouper operates by applying data-parallel processing pipelines[3] to partition these "base" datasets. Notably, Dataset Grouper allows user-specified partition methods, but they must operate in an embarrassingly parallel manner. This decision is a formal trade-off made for scalability reasons. Sequential partitioning (e.g., deciding which group has an example $x$ based on which group has an example $y$) can fail to scale to datasets with billions of examples. Thus, Dataset Grouper supports (at scale) embarrassingly parallelizable partitions of datasets available in TensorFlow Datasets or HuggingFace Datasets.

## 3.3 Compatibility with Existing Frameworks

Foundation model research and FL research also share a common feature in that there is a wide array of available simulation frameworks. Another goal of our work is to support as wide an array of such frameworks as possible. To that end, Dataset Grouper provides access to datasets as nested iterators

---

[2]We use the TFRecord format [70] for all datasets.

[3]We use Apache Beam pipelines, which are also used by TensorFlow Datasets and HuggingFace Datasets.

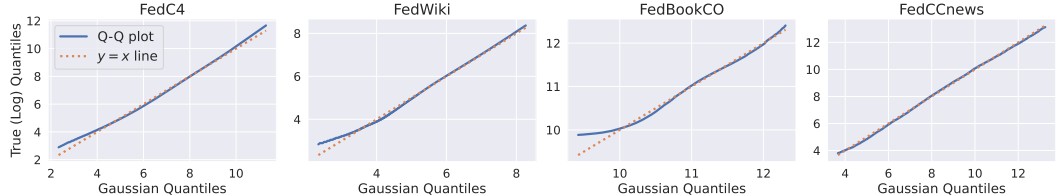

Figure 3: Fitting a log-normal distribution to the per-group sizes of the new text datasets we introduce: we show a Q-Q plot of the log quantiles of the per-group data sizes vs. those of a Gaussian distribution.

of tensors. Specifically, group-structured datasets are represented as an iterator of group datasets, each of which is an iterator of tensors. These tensors can be represented in both TensorFlow [70] and NumPy [72] formats, ensuring that, in principle, Dataset Grouper can be used in any simulation framework built on top of NumPy, TensorFlow, PyTorch [73], or JAX [74].[4]

## 4 Examples and Applications

We now focus on four new group-structured text datasets we create via Dataset Grouper: FedC4, FedWiki, FedCCnews, FedBookCO. We focus on language modeling datasets due to their prominence in training foundation models and their large-scale nature. Compared to prior benchmark datasets, FedC4 is an order of magnitude larger, while FedBookCO contains significantly longer sequences. The new datasets, particularly FedC4 and FedCCnews, are also more heavy-tailed than existing ones. See Appendix B for more details.

While representative of the statistical structure suited to training larger models, we wish to emphasize that these datasets are only a small sample of what is possible with Dataset Grouper. The library can also be used to create group-structured multi-lingual text datasets, datasets in other modalities (audio, image, etc.), and to study the effect of different partitions on the same base dataset.

**FedC4.** We create a federated version of the Colossal Clean Crawled Corpus (C4) dataset [18], which is a cleaned version of Common Crawl's web crawl corpus.[5] We focus on partitioning by web domain, e.g., all articles crawled from https://www.nytimes.com/ correspond to one group. We note that a finer partitioning at the level of articles is also possible. We see from Figure 1 and Table 1 that the amount of data per client is extremely heavy-tailed; this is expected from real-world text corpora [75, 76]. Indeed, this distribution is nearly log-normal, meaning that its logarithm is approximately Gaussian. This can be seen from the nearly straight line in the Q-Q plot in Figure 3.

The C4 data is also used as a pre-training corpus for some LLMs such as T5 [18], meaning that FedC4 can potentially be used for federated pre-training, which we explore further in Section 5. Note that C4 is already de-duplicated and artifacts like code, placeholder text (e.g. lorem ipsum), and boilerplate (e.g. menus) are removed along with a heuristic list of "bad words". See [18] for details.

**FedWiki.** We create a federated version of the Wikipedia dataset, where each client contains the content of one full English Wikipedia article. As a result, the amount of data per client is smaller than that in FedC4, where each client can contain multiple articles. Wikipedia data is often a part of the pre-training corpora of LLMs.

**FedBookCO.** We create a federated version of the BookCorpusOpen dataset [77, 78], an open-source collection of 18K books from various genres. Each client corresponds to one sequence that is a full book, leading to significantly longer sequences than other datasets.

**FedCCnews.** We create a federated version of CC-News, which contains English news articles from news sites around the world. Similar to FedC4, each group corresponds to a web domain; a finer-grained article-level partitioning is also possible. Indeed, FedCCnews is a subset of FedC4. It exhibits similar long-tailed behavior and could serve well as its smaller proxy.

---

[4]Some frameworks are only inter-operable with specific in-memory or hierarchical dataset formats, and would need to be extended to be compatible with Dataset Grouper. Other frameworks are directly compatible. We provide examples of integrating Dataset Grouper with both TensorFlow Federated and JAX.

[5]https://commoncrawl.org/

| Cohort Size | Data Iteration Time (s) | Training Time (s) | Data Iteration Time (%) |
|:---:|:---:|:---:|:---:|
| 8 | $0.26 \pm 0.48$ | $3.03 \pm 2.58$ | 7.78 |
| 16 | $0.66 \pm 0.85$ | $5.70 \pm 2.61$ | 10.43 |
| 32 | $1.16 \pm 1.48$ | $11.30 \pm 2.48$ | 9.33 |

Table 4: Average time spent per round on iterating over data, including preprocessing, versus training. Results are computed for 100 rounds of training of FedAvg, with varying cohort sizes.

## 5 Experiments

To begin to demonstrate the scale of federated learning simulation that these newly partitioned datasets enable, we run experiments on FedC4 with a decoder-only transformer architecture.

### 5.1 Experimental Setup

We use FedC4 with domain-level partitioning in our experiments. We use a WordPiece tokenizer [79] with a pre-trained BERT vocabulary [80] of size of 30523. We train *from scratch* a 108M parameter decoder-only transformer model commensurate in size with BERT base and GPT-2 small (i.e., 12 layers, 12 attention heads, and hidden layers of dimension 768) using the causal language modeling loss (i.e., next token prediction with cross-entropy). We report the cross-entropy loss throughout, which equals the logarithm of the perplexity.

**Federated algorithms.** We use two prototypical FL algorithms: FedAvg and FedSGD [81]. In each federated round, we select the next cohort of 16 clients. Local training is done on client data batched to 16 examples with a sequence length of 128 tokens. We repeat client data as necessary to ensure that all clients have 1024 examples. For FedAvg, we run 3125 rounds of federated training. During each round, each client in that round's cohort takes 64 gradient steps. Thus, the federated training will involve roughly 200K batched gradient computations in total. For FedSGD, we use the same setup, except that clients do not locally update their own model when computing local gradients. Instead, these 64 minibatch gradients are averaged into a single large-batch gradient and sent to the server.

**Optimizer hyperparameters.** For FedAvg, we use the client/server-optimizer framework proposed by Reddi et al. [30]. We use SGD for the client optimizer and Adam for the server optimizer. FedSGD only has a server optimizer, which we also set to Adam. We only tune the learning rate(s), tuning over $\{10^{-4}, 10^{-3}, \ldots, 10^0\}$, and selecting the learning rate(s) that minimize average training loss. For details and a full list of optimizer hyperparameters, see Appendix C.

**Hardware configuration.** We run our experiments using a TPU Pod slice consisting of 16 TPU v3 chips in a 4x4 topology, configured to use a multi-machine inter-chip interconnect mesh. Each TPU v3 chip contains two TensorCores, 32 GiB of high-bandwidth memory, 87.25 GiB RAM, 900 GBps bandwidth, and 123 teraflops peak compute.

### 5.2 Experimental Results

**Iteration efficiency.** We test the efficiency of Dataset Grouper in practical large-scale simulations. Specifically, we measure the time it takes for each round of federated training and what portion of that time is spent iterating over data, including preprocessing. We perform 100 rounds of FedAvg for varying cohort sizes (the number of clients per round) and present the results in Table 4. We see that dataset iteration takes under $10\%$ of the total runtime, even for larger cohort sizes. This is despite the fact that dataset iteration is done entirely on the host, while the training time is parallelized between multiple TPU slices. Further improvements in the data pipeline can only lead to a marginal speedup, highlighting the efficiency and scalability of the streaming dataset format in Section 3.1.

**Federated learning rate schedules.** Large-scale training on non-partitioned data generally involves a variety of important techniques, such as learning rate scheduling, to attain good performance. In order to determine how best to scale federated training to larger-scale settings, we investigate the use of various learning rate schedules for FedAvg and FedSGD. In both cases, we apply the learning rate schedule at the *server* (see [30] for a discussion of client versus server optimizers). We use constant

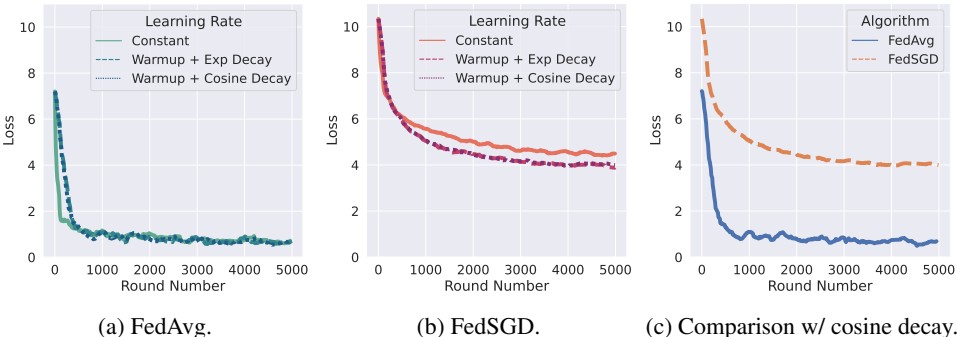

| (a) FedAvg. | (b) FedSGD. | (c) Comparison w/ cosine decay. |

Figure 4: Training loss of FedAvg and FedSGD on FedC4 with different learning rate schedules. The per-round training loss is computed by (a) averaging over all batches seen by a given client within the round, and (b) averaging over all clients that participate in the round.

learning rates, warmup with exponential decay, and warmup with cosine decay. Whenever we use warmup, we warmup for 10% of the total number of training rounds, and decay for the remainder.

We compare the resulting training loss for FedAvg and FedSGD in Figure 4. Notably, we see that learning rate scheduling leads to significant improvements in the behavior of FedSGD, while FedAvg is robust to different choices. This reflects the fact that these learning rate schedules were developed in the context of SGD, which involves applying many unbiased gradient estimates. FedSGD operates similarly, computing an unbiased gradient estimate at each round. By contrast, FedAvg involves biased gradients, often called "pseudo-gradients" [30], which may not be the gradient of any loss function [82]. Our results suggest that developing effective learning rate scheduling techniques for FedAvg is an open question, and may involve coordinating client and server learning rates.

We also see that FedAvg appears to attain a significantly lower train loss than FedSGD. We stress that this is due to how the training loss is computed. For both algorithms, it is computed by averaging the loss of all batches seen by a client and then averaging that quantity across all clients. However, the client trains as it sees data batches in FedAvg. Therefore, the client's local model adapts to its own distribution (leading to a lower loss), while in FedSGD the client does not adapt its local model. We explore this difference, which is connected to **meta-learning**, below.

**Federated evaluation and personalization.** Partitioned datasets enable group-structured learning, as well as group-level (or federated) evaluation, which may be particularly informative for measuring downstream performance across heterogeneous data splits. To demonstrate this, we use Dataset Grouper to generate an evaluation dataset from FedC4 by using its held-out validation split. We use the same partition structure as before, grouping examples according to their base domain. Because of this group structure, we can compute histograms of metrics across all groups, rather than just an average metric across all examples.

We take the resulting models trained by FedAvg and FedSGD (with constant learning rates, though we see similar results for all learning rate schedules we considered above), and compute two separate metrics for each validation client. First, we compute the average loss of the model on all examples held by the client. We refer to this as the **pre-personalization loss**. We then fine-tune the model for a single epoch on the client's dataset (using a client optimizer of SGD with a tuned learning rate). After personalization, we compute the average loss again, resulting in the **post-personalization loss**.

| Algorithm | Pre-Personalization Loss | | | Post-Personalization Loss | | |
|---|---|---|---|---|---|---|
| | 10th perc. | Median | 90th perc. | 10th perc. | Median | 90th perc. |
| FedAvg | 5.13 | 5.64 | 6.27 | **0.002** | **0.012** | **0.934** |
| FedSGD | **4.38** | **4.93** | **5.40** | 1.25 | 3.38 | 4.53 |

Table 5: Validation loss of FedAvg and FedSGD, before and after personalizing on a client's dataset. Percentiles are computed across all clients in the FedC4 validation dataset.

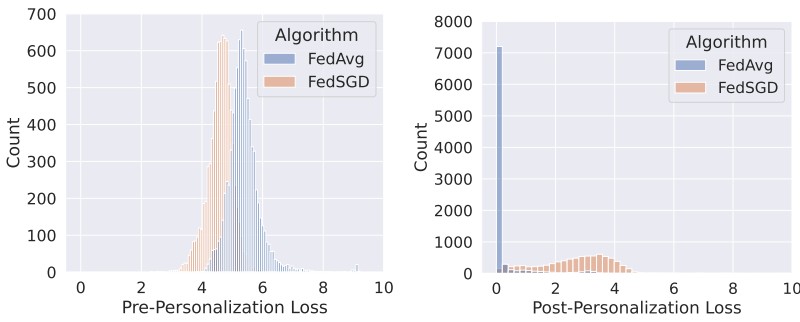

Figure 5: Histograms of pre- and post-personalization loss across all FedC4 validation clients.

We present quantiles of these metrics in Table 5. Intriguingly, they show that the FedSGD-trained model works better for pre-personalization, but the FedAvg-trained model is much more effective at personalizing to the client's data. To further illustrate this, we consider histograms of the two distributions (across all clients) in Figure 5. This suggests a more dramatic shift. While the FedAvg- and FedSGD-trained models are close in pre-personalization performance (though FedSGD does better), the post-personalization distribution for FedAvg is extremely light-tailed.

**Task-specific personalization.** Pre-trained foundation models are typically employed on a range of downstream tasks. In this spirit, we use the models trained on FedC4 to perform pre- and post-personalization evaluation on FedBookCO. The results, Figures 6 and 7, are similar to but less drastic than those of Figure 5. The pre-personalization loss of FedAvg is slightly larger than FedSGD (5.0 vs. 4.3 in the last checkpoint) while its post-personalization loss is smaller (2.9 vs. 4.0). Similar trends hold for FedCCnews and FedWiki datasets; cf. Appendix D. Overall, these results show that FedAvg's superior personalization performance is **robust to shifts in the distribution over clients**.

This phenomenon highlights connections between federated learning and meta-learning previously noted in the literature [46, 83–88]. In short, we see that FedAvg acts as a meta-learning algorithm (specifically, the Reptile algorithm [89]) where it quickly minimizes the loss of a client after a few local gradient steps (i.e., after personalization). It does not behave like an algorithm designed to minimize the empirical risk. By contrast, FedSGD operates much like SGD in the centralized setting, attempting to minimize the average loss across all examples. To the best of our knowledge, Figure 5 constitutes some of the strongest empirical evidence of the connection between federated and meta-learning to date. The scale of the FedC4 dataset (enabled by Dataset Grouper) is critical here, as clients have sufficiently large amounts of data to exacerbate client drift [28] and cause tension between loss minimization and meta-learning.

**Scaling to larger models.** To further demonstrate the scalability of Dataset Grouper, we train a transformer model with 1 billion parameters on the FedC4 dataset. In contrast to the results above, we train with 4 batches per client (rather than 64). Despite this, we see in Figure 8 that FedSGD still sees

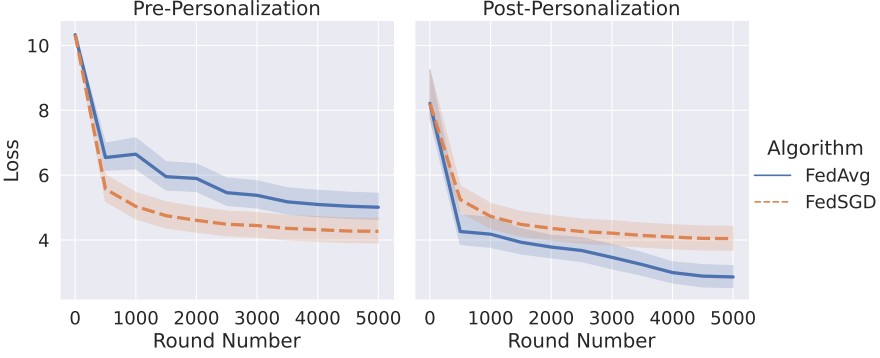

Figure 6: Median pre-personalization (left) and post-personalization (left) loss over FedBookCO clients while training on FedC4. The shaded region indicates the 10th and 90th percentiles.

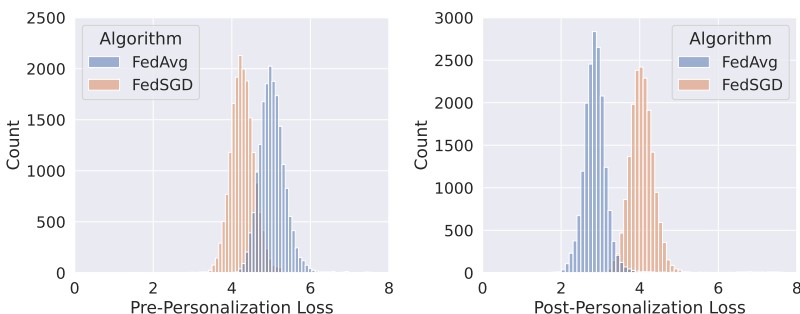

Figure 7: Histograms of pre- and post-personalization loss on FedBookCO after FedC4 training.

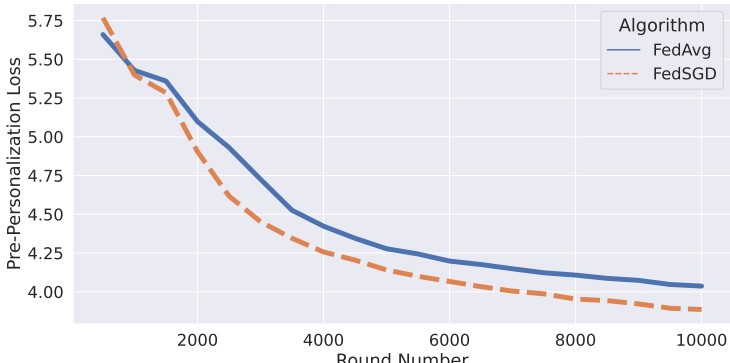

Figure 8: Pre-personalization loss of FedAvg and FedSGD across all FedC4 validation clients on a 1 billion parameter transformer model.

improved pre-personalization loss compared to FedAvg. Moreover, both algorithms see improved pre-personalization compared to Table 5, highlighting the effect of increasing the larger size.

## 6    Discussion and Outlook

The intersection of foundation models and group-structured data is a fertile area for research. We provide tooling for creating group-partitioned datasets for use in large-scale research simulation. We acknowledge that there are inherent risks in this endeavor. As detailed by Koch et al. [90], the typical dynamics of dataset use in machine learning research tend to enshrine certain datasets as "sole benchmarks" in the field, even on tasks for which they were not designed. Tooling aimed at allowing for flexible and reproducible dataset creation risks further entrenchment of these sole benchmarks by expanding the scope of tasks to which they are applied. However, we posit that Dataset Grouper's pipeline approach will prove to be a sustainable mechanism for ensuring availability of datasets whose intended use cases match their application in research, and can potentially reduce the enshrinement of benchmarks in areas such as federated learning.

There are a wide array of other research benefits enabled by Dataset Grouper, especially as it delivers scalable and efficient partitioned dataset pipelines compatible with foundation model training and fine-tuning. This crucially enables the exploration of phenomena that only emerge at the scale of foundation models. Our experiments are intended as a demonstration of the scaling capabilities unlocked by Dataset Grouper to the billion parameter regime. Our empirical findings indicate several interesting future directions. Most excitingly (and speculatively), the tendency of FedAvg to meta-learn tasks suggests that it could provide a better "base" model for the personalization of foundation models or adaptation to downstream tasks. Moreover, there is a need to design tailored learning rates and default optimization recipes for the wider applicability of federated training algorithms. We hope that Dataset Grouper will spur further research in training, evaluation, finer grained analyses, and diagnoses of foundation models with group-structured data.

**Acknowledgements**

The authors thank Keith Rush, H. Brendan McMahan, Sean Augenstein, and Liam Collins for fruitful discussions and helpful comments. The authors would also like to thank Keith Rush for help with training code and multi-TPU support. Finally, the authors would like to thank Zheng Xu, Shanshan Wu, Keith Rush, and Arun Ganesh for helpful conversations on group-structured datasets.

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

# Appendix

## Table of Contents

# A Software

All code for Dataset Grouper can be found on GitHub,[6] including all applicable licenses, disclaimers, usage instructions, and examples. We showcase some functionalities below.

## A.1 Installation, Usage and Examples

**Installation.** Dataset Grouper can be installed as a Python package[7] using the pip command `pip install dataset-grouper`.

**Basic usage.** At its core, Dataset Grouper can be used to partition datasets (eg. from TensorFlow Datasets [16]) across groups. It can be used with any embarrassingly parallel user-defined partitioning function of the signature `get_key_fn(example) -> group_id`.

Below we give a simple example where we partition the MNIST dataset according to label (so that we form one group per label).

Listing 1: Using Dataset Grouper to partition MNIST

```python
import apache_beam as beam
import dataset_grouper as dsgp
import tensorflow_datasets as tfds

# First we download the MNIST dataset from TFDS.
dataset_builder = tfds.builder("mnist")
dataset_builder.download_and_prepare(...)

# Next, we define a function that maps MNIST examples to their group.
def get_label_fn(x):
  label = x["label"].numpy()
  return str(label).encode("utf-8")

# Finally, we build the partitioning pipeline, and run it using Apache Beam.
mnist_pipeline = dsgp.tfds_to_tfrecords(
    dataset_builder=dataset_builder,
    split="train",
    get_key_fn=get_label_fn,
    file_path_prefix=...
)
with beam.Pipeline() as root:
  mnist_pipeline(root)
```

**Dataset partitioning examples.** We give additional examples of using Dataset Grouper to partition datasets into groups:

- Feature-based partitioning: This script allows partitioning of a dataset by one of its features, allowing for natural heterogeneous partitions. The language modeling datasets in Section 4 use this approach. For example, to create FedCCnews and FedC4 we partition based on the URL.

- Random partitioning: This script assigns each example to client at random.

- Heterogeneous partitioning using a Dirichlet process: This script assigns each example to a client based on a Dirichlet process. This is an embarrassingly parallel version of the popular LDA-based method that is popular in the federated learning literature [e.g. 71]. Together with the random partitioning, this can be used, for instance, to study the effect of the level of data heterogeneity on FL algorithms.

The specific commands used to create the datasets introduced in Section 4 can be found in the GitHub repository at `https://github.com/google-research/dataset_grouper/tree/main/dataset_grouper/examples/datasets`.

---

[6]`https://github.com/google-research/dataset_grouper`
[7]`https://pypi.org/project/dataset-grouper/`

**Usage in training loops.** Dataset Grouper saves partitioned datasets to the widely used TFRecord format and builds grouped dataset pipelines via `tf.data` [91]. It can be used as shown below.

Listing 2: Iterating over group-partitioned datasets via Dataset Grouper

```
1  import dataset_grouper as dsgp
2
3  # Load the dataset by passing the file. paths and the TFDS dataset name
4  partitioned_dataset = dsgp.PartitionedDataset(
5          file_pattern=...,
6          tfds_features="c4" # Or any other TFDS dataset name.
7  )
8  # Obtain an iterator of clients/groups.
9  client_stream = partitioned_dataset.build_group_stream()
10 # Iterate over clients.
11 for client_dataset in client_stream:
12     # client_dataset is an iterable of examples.
13     for example in client_dataset.as_numpy_iterator():
14         # Process this example.
```

Typically, FL processes cohorts of clients. This can easily by achieved by applying a `batch` operation on the `client_stream` object above; see the training code below for details.

**Training code examples.** The GitHub repository contains JAX code to reproduce the experiments discussed in Section 5 and Appendix C. It also gives sample code to demonstrate the integration of Dataset Grouper with TensorFlow Federated [11].

## A.2 Dataset Hosting

Dataset Grouper does not directly host any datasets and instead provides tools for downloading, preparing, and partitioning publicly available datasets. As with most design decisions, this is a trade-off. It enables Dataset Grouper to focus on utilities that apply to a broad range of datasets (notably, all datasets hosted by TensorFlow Datasets [16] and HuggingFace Datasets [17]). However, it also means that we cannot guarantee the hosting of various datasets in perpetuity. That being said, the tools underlying Dataset Grouper generalize to a wide array of settings (namely, any setting in which the base dataset can be accessed via an Apache Beam pipeline), and as such we believe Dataset Grouper provides a concrete benefit to machine learning researchers and practitioners working with group-structured data.

## A.3 Dataset Licenses

Before using Dataset Grouper to partition a "base" dataset, users should also ensure that their use case falls under the license of that base dataset and that they abide by the associated terms and services. For example, below, we discuss the end license of the four (non-partitioned) datasets we use to create the partitioned datasets in Section 4.

**BookCorpusOpen.** We access this dataset through HuggingFace datasets. This dataset was originally derived from smashwords.com, and therefore should be used in compliance with the associated terms of service. We encourage readers to read the datasheet for BookCorpus created by Bandy and Vincent [78]. This datasheet identifies potential deficiencies of the dataset, including problematic and skewed content, possible copyright issues, and book duplication.

**CC-News.** We access this dataset through HuggingFace datasets. This dataset is publicly available and hosted by https://commoncrawl.org/, and therefore should be used in compliance with the associated terms of use.

**C4.** We access this dataset through TensorFlow Datasets. This dataset is based on a publicly available dataset hosted by https://commoncrawl.org/ and therefore should be used in compliance with the associated terms of use.

**Wikipedia.** We access this dataset through TensorFlow Datasets. This dataset is created through Wikimedia downloads (https://dumps.wikimedia.org/). Each example is derived from single Wikipedia articles and subjected to post-processing and cleaning to strip markdown and unwanted sections (references, etc.). This dataset should be used in compliance with Wikipedia's terms of use.

## A.4 Software Development and Maintenance

Dataset Grouper is actively developed and maintained using standard open-source workflows. In particular, we accept responsibility for reviewing and acting on issues and contributions from the community, as long as they abide by a contributor license agreement discussed in the repository.

# B   Dataset Statistics

| Source | Dataset | Partition on | #Clients | #Words | #Words per client | | | | |
|---|---|---|---|---|---|---|---|---|---|
| | | | | | 10th perc. | 25th perc. | Median | 75th perc. | 90th perc. |
| **Ours** | FedC4 | Domain | 15.6M | 132B | 82 | 220 | 815 | 3.3K | 0.5B |
| | FedWiki | Article | 6.5M | 3B | 39 | 75 | 198 | 486 | 70K |
| | FedBookCO | Book | 18K | 1.2B | 24K | 32K | 52K | 81K | 4M |
| | FedCCnews | Domain | 8.8K | 0.3B | 303 | 1.1K | 5K | 20K | 8.4M |
| **Existing** | Amazon Reviews | Account | 1.5M | 4.3B | 278 | 565 | 1.1K | 2.3K | 5K |
| | Stack Overflow | Account | 0.3M | 2B | 1.2K | 1.7K | 2.7K | 5.1K | 11K |
| | Reddit | Account | 1.7M | 1.2B | 58 | 111 | 257 | 675 | 1720 |
| | Blog Corpus | Account | 17K | 0.1B | 551 | 908 | 2K | 5.3K | 13K |
| | Shakespeare | Role/play | 715 | 0.4M | 14 | 45 | 175 | 0.6K | 1.6K |
| | Gigaword | Synthetic | 100 | 0.3M | 3.0K | 3.1K | 3.1K | 3.2K | 3.2K |

Table 6: Detailed version of Table 1: A summary of the per-client statistics of the new language modeling datasets we introduce using Dataset Grouper, and of previous benchmark datasets supplied by TFF, Leaf, FedNLP, and FedScale.

| Source | Dataset | Partition on | #Examples | #Words | #Words per Exmaple | | | | |
|---|---|---|---|---|---|---|---|---|---|
| | | | | | 10th perc. | 25th perc. | Median | 75th perc. | 90th perc. |
| **Ours** | FedC4 | Domain | 0.36B | 132B | 49 | 88 | 191 | 417 | 783 |
| | FedWiki | Article | 6.5M | 3B | 39 | 75 | 198 | 486 | 70K |
| | FedBookCO | Book | 18K | 1.2B | 24K | 32K | 52K | 81K | 4M |
| | FedCCnews | Domain | 0.7M | 0.3B | 78 | 151 | 316 | 548 | 842 |
| **Existing** | Amazon Reviews | Account | 68M | 4.3B | 3 | 10 | 28 | 67 | 155 |
| | Stack Overflow | Account | 0.1B | 2B | 3 | 7 | 13 | 20 | 29 |
| | Reddit | Account | 33M | 1.2B | 7 | 11 | 21 | 42 | 81 |
| | Blog Corpus | Account | 0.5M | 0.1B | 6 | 28 | 105 | 248 | 460 |
| | Shakespeare | Role/play | 16K | 0.4M | 4 | 8 | 12 | 29 | 63 |
| | Gigaword | Synthetic | 10K | 0.3M | 21 | 26 | 31 | 36 | 41 |

Table 7: Detailed version of Table 1: A summary of the per-example (i.e., per-sequence) statistics of the new datasets we introduce using Dataset Grouper, and of previous benchmark datasets supplied by TFF, Leaf, FedNLP, and FedScale.

Here, we detail various statistics of the federated language-modeling datasets we propose and discuss in Section 4. In Table 6, we present the per-client statistics of the dataset, and compare these to per-client statistics of existing federated language modeling datasets supplied by TensorFlow Federated [11], Leaf [12], FedNLP [13], and FedScale [15]. We see that at larger percentiles, FedC4, FedWiki, FedBookCO, and FedCCnews contain many more words per client. FedBookCO also contains dramatically more words per client at lower percentiles than previous datasets. In general, datasets like FedC4 and FedCCnews exhibit a dramatic variance in statistics across clients. This can make the datasets more challenging for federated algorithms, and potentially more representative of heavy-tailed settings in practice.

In Table 7, we compare the per-example (i.e. per-sequence) statistics of the same datasets. We note that this information is important for large-scale language modeling tasks, which may require long sequences of tokens for training. We see that for nearly every percentile, the datasets we introduce contain sequences of significantly larger lengths. This is especially true of larger percentiles, at which point FedBookCO and FedWiki contains examples with thousands if not millions of words.

To better visualize and compare the distribution of words per client, and given the large number of clients in each dataset, we present a letter value plot [92] of these distributions. The result is in Figure 9. This plot gives various quantiles of the distribution of the number of words per client. We see that many of these datasets exhibit large amounts of variance between quantiles. FedC4 has an especially heavy tail, with some clients having fewer than 10 words, while others have tens or even hundreds of millions of words.

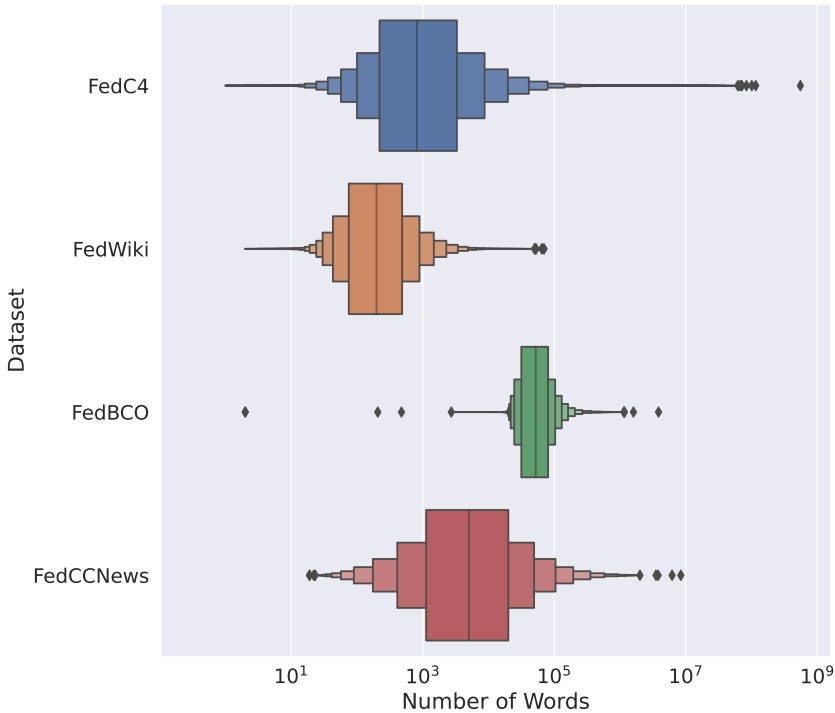

Figure 9: A letter value plot of the number of words held by each client, in the FedC4, FedWiki, FedBookCO, and FedCCnews datasets.

**Example application scenarios.** FedBookCO (each client is a book) and FedWiki (each client is a Wikipedia article) map to applications where each client is an "expert" in a certain topic. In the context of modern LLM pipelines where each sequence is broken up into multiple examples of a fixed length, the fact that each client has a single sequence is much less important than the total number of words. This is especially true for FedBookCO where the $10^{th}$ percentile data sequence length is 24K words, while the maximum sequence length for LLMs today is $O(1K)$ words.

FedC4 is typical of a group-structured pre-training (or second stage pre-training) dataset, which might be practically encountered in practice with documents (corporate, medical, legal, etc.) or emails. FedCCnews is a subset of FedC4 with similar long-tailed characteristics and is suitable for faster experimentation of second stage pre-training or fine-tuning. It can also be used to study the effects of pretraining data contamination.

## C   Full Experimental Details

We describe the full details of the experiments in Section 5. We discuss the datasets and preprocessing in Appendix C.1, the model in Appendix C.2, the federated algorithms in Appendix C.3, the hyperparameter choices in Appendix C.4, the reported metrics in Appendix C.5, and the hardware configuration in Appendix C.6.

## C.1 Datasets and Preprocessing

We use FedC4 with domain-level partitioning for all experiments, which we discuss in Section 4. We use a WordPiece tokenizer [79] with a pre-trained BERT vocabulary [80] of size 30523. For each client, we concatenate all of the text in its examples into sequences of tokens of length 129, padding the last sequence as needed. In each sequence $x_{1:129}$, we predict $x_2$ given $x_1$, $x_3$ given $x_{1:2}$ and so on until $x_{129}$ given $x_{1:128}$. This leads to a total of 128 predictions per sequence. We batch the sequences with a batch size of 16 and apply "take" and "repeat" operations to ensure that each client has exactly 64 batches. For evaluation, we follow the same procedure. However, we use the "validation" split of the C4 dataset, rather than the "train" split.

## C.2 Model

We use a decoder-only transformer model whose size is roughly on the order of the BERT base or GPT-2 small. It has 12 layers, 12 attention heads, and hidden layers of dimension 768. As discussed above, it makes 128 predictions in each sequence using the causal language modeling loss (i.e. next-token prediction with cross-entropy loss).

## C.3 Federated Algorithms

We use two federated learning algorithms, FedAvg and FedSGD [81]. In both algorithms, at each round $t$ the server model $x^t$ is sent to all the clients participating in round $t$ (i.e. the cohort at round $t$). Throughout, 16 clients participate in each round (i.e. the *cohort size* is 16). As discussed in Section 3, we shuffle the clients globally once and iterate successively through the stream of shuffled clients in windows of size 16. In both algorithms, each client uses this model to compute a gradient at each of its 64 batches. However, the algorithms differ in where these gradients are computed.

For FedAvg, we actually use the slightly generalized FedOpt framework [30] in which FedAvg uses both a client optimizer and a server optimizer. Throughout, we use SGD as the client optimizer and Adam as the server optimizer. After a client $c$ computes a gradient, it updates its model locally using the client optimizer (i.e. SGD), starting at the broadcast model $x^t$. After $K = 64$ updates, this results in some updated model $x_c^t$. The client then sends $\Delta_c^t := x^t - x_c^t$ to the server. For FedSGD, the clients do not locally update their model. Each gradient is computed at the broadcast model $x^t$, and each client $c$ sends the average $\Delta_c^t$ of these gradients to the server.

Once the server has received $\Delta_c^t$ for each participating client, it averages them uniformly (as weighted and uniform are the same in our setting) to produce some quantity $\Delta^t$. The server treats this as an estimate of the gradient of the empirical risk function at the model $x^t$ and applies the server optimizer (i.e. Adam) to update the model accordingly.

For both algorithms, we perform 3125 rounds of federated training. This means that the server model is updated 3125 times and that throughout the course of training, $3125 \times 16 \times 64 = 3.2\text{M}$ batched gradients are computed (as each client has 64 batches, and 16 clients participate in each round). If we consider each client to be computing gradients simultaneously, and form "meta-batches" of gradients of size 16 over the batched gradients, this means that we are doing roughly $3125 \times 64 = 200\text{K}$ "meta-batched" gradient computations in total.

Note that when doing personalization for the purposes of evaluation, as in Table 5 and Figure 5, we personalize the model (trained via FedAvg or FedSGD) using the same client training scheme as in FedAvg: Clients perform 64 steps of SGD on the model broadcast to them. The batches of data are formed in the exact same way as in training.

## C.4 Optimizer Hyperparameters

As discussed in Appendix C.3, for both FedAvg and FedSGD we use a server optimizer of Adam. We tune only the learning rate of this optimizer (the *server learning rate*, denoted $\eta_s$) and fix the Adam hyperparameters of $\beta_1 = 0.9, \beta_2 = 0.999$ and $\varepsilon = 10^{-8}$. For FedAvg we also use a client optimizer of SGD. We tune the learning rate of this optimizer (the *client learning rate*, denoted $\eta_c$). We tune both learning rates $\eta_s, \eta_c$ over the range $\{10^{-4}, 10^{-3}, \ldots, 10^0\}$. We select the learning rates that minimize average training loss across rounds. See Table 8 for a summary.

| Placement | Optimizer | Hyperparameter | Value | Tuning Range |
|---|---|---|---|---|
| Server | Adam | Learning Rate ($\eta_s$) | N/A | $\{10^{-4}, 10^{-3}, \ldots, 10^0\}$ |
| | | First Moment Decay ($\beta_1$) | 0.9 | N/A |
| | | Second Moment Decay ($\beta_2$) | 0.999 | N/A |
| | | Numerical Stability Term ($\varepsilon$) | $10^{-8}$ | N/A |
| Clients | SGD | Learning Rate ($\eta_c$) | N/A | $\{10^{-4}, 10^{-3}, \ldots, 10^0\}$ |

Table 8: Optimizer hyperparameters.

| Algorithm | Server LR Schedule | Server LR ($\eta_s$) | Client LR ($\eta_c$) |
|---|---|---|---|
| FedAvg | Constant | $10^{-3}$ | $10^{-1}$ |
| | Warmup + Exponential Decay | $10^{-3}$ | $10^{-1}$ |
| | Warmup + Cosine Decay | $10^{-3}$ | $10^{-1}$ |
| FedSGD | Constant | $10^{-4}$ | N/A |
| | Warmup + Exponential Decay | $10^{-3}$ | N/A |
| | Warmup + Cosine Decay | $10^{-3}$ | N/A |

Table 9: Tuned learning rates for FedAvg and FedSGD, with varying server learning rate schedules.

In Figure 4 we also experiment with learning rate scheduling. Note that these are only applied to the server optimizer. The client learning rate (for FedAvg) is held constant throughout an experiment. The learning rate schedule is applied across the 3125 training rounds. We compare constant learning rates to learning rates with (1) exponential decay and (2) cosine decay. For both of these decay schedules, we perform linear warmup (starting at 0) for the first 312 rounds ( 10% of the total number of rounds). We then decay for the remaining rounds, with a final server learning rate of 0. In such cases, the server learning rate parameter $\eta_s$ refers to the *maximum* learning rate attained (i.e. at round 312) and is tuned just as above.

The best performing (tuned) learning rates for each algorithm and schedule are given in Table 9. Generally, we found that a client learning rate of $\eta_c = 10^{-1}$ worked well throughout. For FedAvg, a server learning rate of $\eta_s = 10^{-3}$ worked well, though we see little to no difference between server learning rate schedules Figure 4a. For FedSGD, we find that we could only use $\eta_s = 10^{-4}$ for constant learning rates, but learning rate schedules allowed us to use $\eta_s = 10^{-3}$ and led to improved convergence Figure 4b.

When performing personalization evaluation (Table 5 and Figure 5), we use the same client optimizer of SGD, and use the client learning rate that led to the best training performance for FedAvg (i.e. $\eta_c = 10^{-1}$, as in Table 9).

## C.5 Reported Metrics

In Figure 4, we report a causal language modeling loss (i.e. the logarithm of the perplexity) at each training round of FedAvg and FedSGD. How we compute these averages depends on the federated algorithm used. For FedSGD, each client computes gradients and loss values across 64 batches, at the same model. The client averages the loss across these 64 batches and sends the result to the server. The server averages these quantities across the 16 clients participating in a round.

We do the same thing for FedAvg, except we must keep in mind that for each client, the 64 different loss values are computed at *different models*. This is because in FedAvg, each client locally updates its model as it computes gradients. The average of these 64 loss values is still sent to the server and averaged across the 16 clients participating in a round. However, because of the local training, this loss represents a different quantity. In short, it accounts for both how good the broadcast model is, and how well the model adapts to a client's data. By contrast, the loss reported for FedSGD only represents how good the broadcast model is.

In Table 5 and Figure 5, there is no such ambiguity. The pre-personalization loss is the average causal language modeling loss *before* a client personalizes the model to its own data. The post-personalization loss is the average causal language modeling loss *after* a client personalizes to its own data.

## C.6    Hardware Configuration

We run our experiments using a TPU Pod slice consisting of 16 TPU v3 chips in a 4x4 topology, configured to use a multi-machine inter-chip interconnect mesh. Each TPU v3 chip contains two TensorCores, 32 GiB of high-bandwidth memory, 87.25 GiB RAM, 900 GBps bandwidth, and 123 teraflops peak compute.

# D    Additional Experimental Results

Here, we present detailed personalization evaluation results on FedCCnews and FedWiki in Appendix D.1, followed by detailed ablation results in Appendix D.2.

## D.1    Personalization Results

Here we present the results of performing pre-personalization and post-personalization evaluation on the FedC4-trained models, but using other datasets. In addition to the evaluation on FedBookCO presented in Section 5, we present results on FedCCnews (Figures 10 and 11), and FedWiki (Figures 12 and 13).

For FedBookCO and FedCCnews, we report the personalization results on the entire dataset. Due to the large number of clients in FedWiki, we randomly sample 20K clients in the dataset for personalization evaluation.

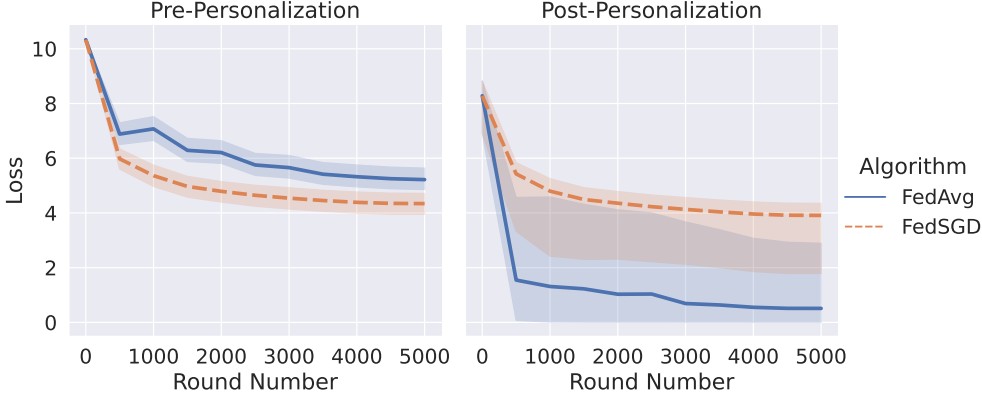

Figure 10: Median pre-personalization (left) and post-personalization (right) loss over FedCCnews clients while training on FedC4. Error bars indicate the 10th and 90th percentiles.

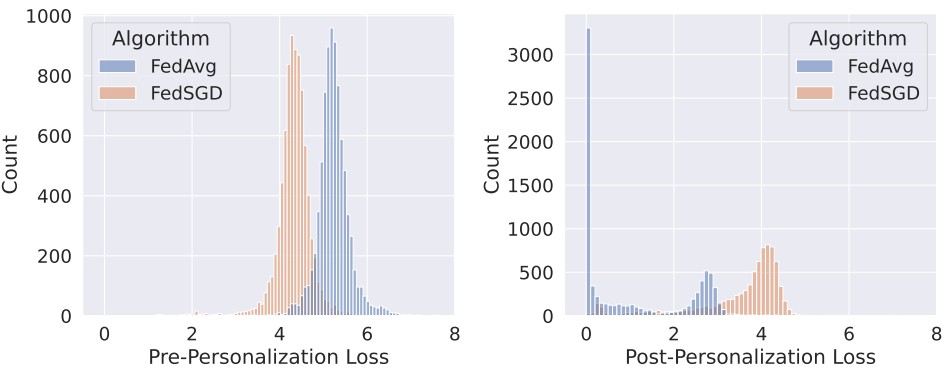

Figure 11: Histograms of pre-personalization (left) and post-personalization (right) loss on FedCCnews after FedC4 training.

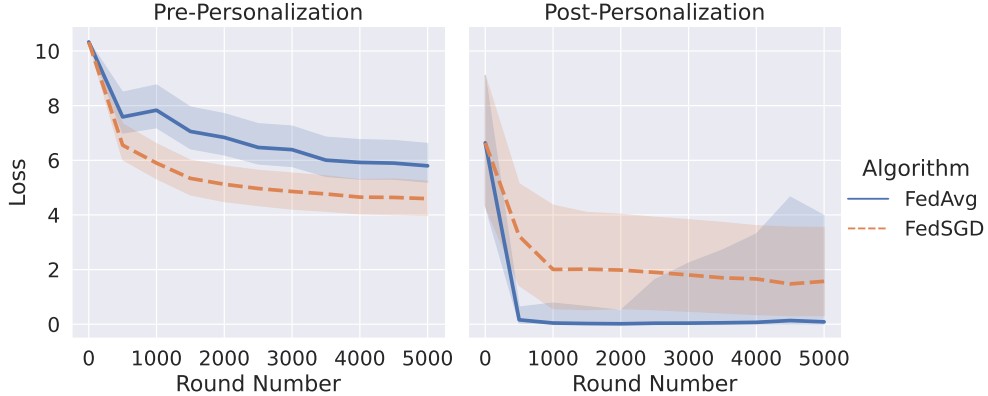

Figure 12: Median pre-personalization (left) and post-personalization (right) loss over FedWiki clients while training on FedC4. Error bars indicate the 10th and 90th percentiles.

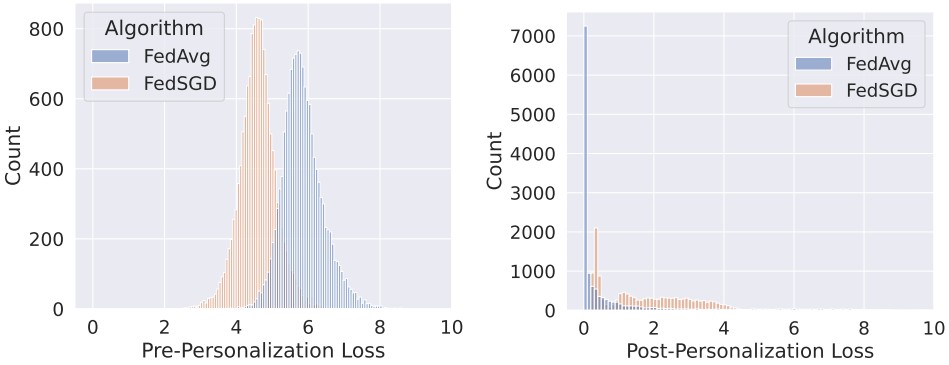

Figure 13: Histograms of pre-personalization (left) and post-personalization (right) loss on FedWiki after FedC4 training.

## D.2 Ablation: Batches per Client

To better explore the phenomena discussed in Section 5, we perform a modified experiment where we repeat the training from Section 5, but vary the number of batches $\tau$ each training client yields. As above, we repeat and truncate clients' datasets so that each client has exactly 1024 examples. For example, when $\tau = 64$, and we use a batch size of 16, this means that for FedAvg and FedSGD, each client computes $\tau = 64$ mini-batch gradients. For FedAvg, this means that the client does $\tau = 64$ steps of training, while for FedSGD, this means that the client sends the average of these $\tau = 64$ batches back to the server. We vary the number of batches $\tau$ over $\{1, 4, 16, 64\}$. We do so by repeating and truncating clients' data so that they have 16, 64, 256, and 1024 examples, respectively. We then perform different amounts of training, one in which we equalize the number of communication rounds, and one in which we equalize the total number of tokens seen across all clients.

Note that fixing the number of examples per client at 1024 and simply changing the batch size would be a more elegant way to do this kind of ablation, as it would allow normalizing the number of communication rounds and tokens simultaneously. Unfortunately, the batch size is often dictated by compute constraints (e.g. what fits in the memory of a given hardware device), and cannot be increased in an unbounded fashion in realistic machine learning settings, especially on-device settings which are common in FL. Thus, we instead focus on varying $\tau$.

**Equalizing communication rounds.** We do 5000 rounds of training for each number of batches per client $\tau$, using the same warmup and cosine decay schedule discussed above. Throughout training, we perform personalization evaluation on the models. We give the pre-personalization results in Figure 14, and the post-personalization results in Figure 15. For more detailed numerical results, see Appendix D.2.

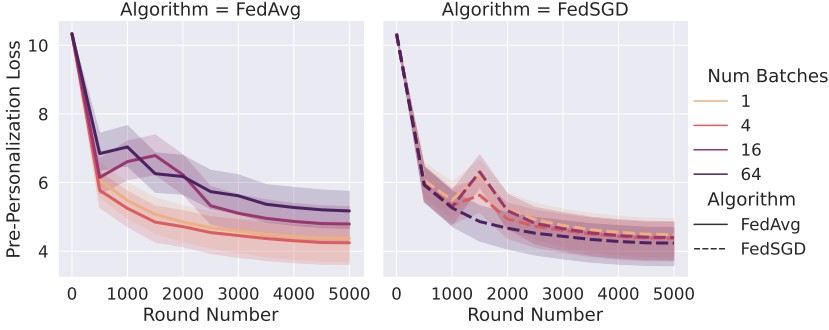

Figure 14: Median pre-personalization loss across FedC4 validation clients, with different numbers of batches used in each client's local computation. Error bars indicate the 10th and 90th percentiles. All runs are equalized to perform the same number of communication rounds in total.

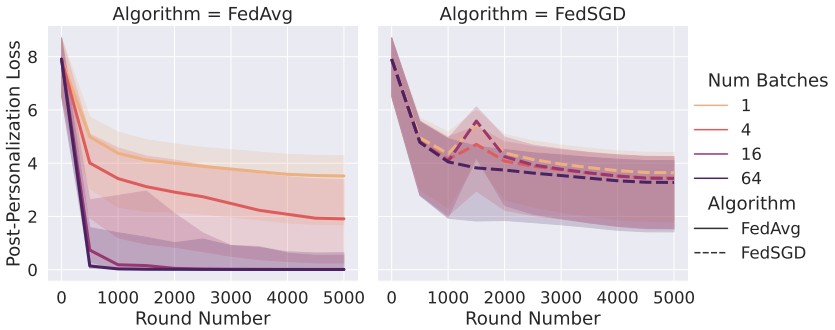

Figure 15: Median post-personalization loss across FedC4 validation clients, with different numbers of batches used in each client's local computation. Error bars indicate the 10th and 90th percentiles. All runs are equalized to perform the same number of communication rounds in total.

We see that for FedSGD, the pre- and post-personalization losses do not change much with the number of batches per client $\tau$. For FedAvg, on the other hand, lower values of $\tau$ attain better

pre-personalization loss, while higher values achieve better post-personalization loss. Note that when the number of batches per client is $\tau = 1$, FedAvg and FedSGD are effectively the same algorithm (up to differences in normalization), and perform only a single update step per client each round. Notably, *FedAvg attains a better trade-off between the pre- and post-personalization metrics*. Specifically, FedAvg with $\tau = 4$ can match the best pre-personalization performance of FedSGD (FedAvg with $\tau = 4$ and FedSGD with $\tau = 64$ both attain a median pre-personalization loss of 4.2), while performing significantly better on the post-personalization metrics (a median loss of 1.9 for FedAvg with $\tau = 4$ versus 3.4 for FedSGD with $\tau = 64$). In short, these results seem to suggest that "client drift" [28] is less of an impediment to federated learning, and more indicative of a trade-off between minimizing pre- and post-personalization loss functions.

| Algorithm | Loss | Batches per Client ($\tau$) | | | |
|---|---|---|---|---|---|
| | | 1 | 4 | 16 | 64 |
| FedAvg | Pre-Personalization | 4.4 | 4.2 | 4.8 | 5.2 |
| | Post-Personalization | 3.5 | 1.9 | 0.009 | 0.008 |
| FedSGD | Pre-Personalization | 4.5 | 4.4 | 4.4 | 4.2 |
| | Post-Personalization | 3.6 | 3.4 | 3.4 | 3.3 |

Table 10: Median pre-personalization and post-personalization loss after training with FedAvg and FedSGD, with different numbers of batches per client, keeping the total number of communication rounds constant.

**Equalizing tokens.** Next, we perform an analogous experiment, but where each setting of $\tau$ (the number of batches per client), is trained for a different number of rounds, so that in each setting the same number of tokens is processed (over all clients). For example, $\tau = 1$ trains for $64\times$ more communication rounds than for $\tau = 64$. Each training run processes roughly 10.5 billion tokens in total. We give the pre-personalization results in Figure 16, and the post-personalization results in Figure 17. For more detailed numerical results, see Appendix D.2. Notably, the post-personalization loss diverged at intermediate stages for $\tau = 1$, but recovered (albeit with high variance across clients) afterwards.

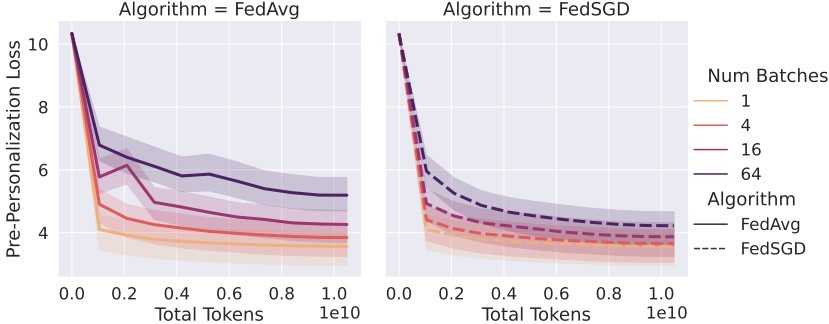

Figure 16: Median pre-personalization loss across FedC4 validation clients, with different numbers of batches used in each client's local computation. Error bars indicate the 10th and 90th percentiles. All runs are equalized to process the same number of tokens in total.

Here, we see a notably different story than in Figures 14 and 15. In particular, we see that for both FedAvg and FedSGD, lower values of $\tau$ lead to lower pre-personalization loss. However, for both algorithms as long as $\tau$ is sufficiently large (at least 4 in this case), the post-personalization loss essentially does not change with $\tau$. This suggests that when communication is not a bottleneck, we can attain good pre- and post-personalization loss by using FedAvg with a small (but not too small) value of $\tau$. In other words, by performing enough rounds of FedAvg with a moderate $\tau$, we can attain a model that does well before and after personalization.

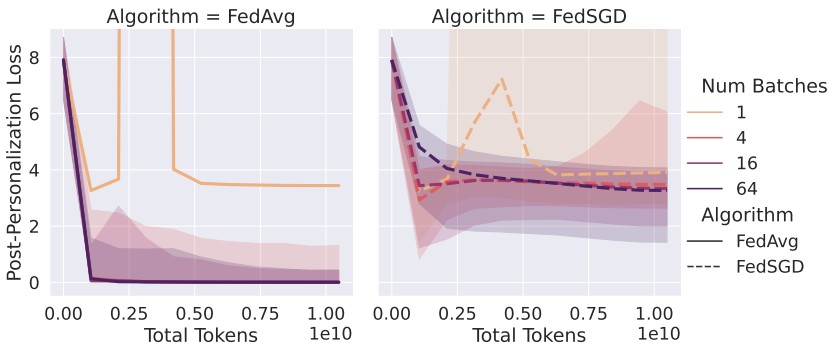

Figure 17: Median post-personalization loss across FedC4 validation clients, with different numbers of batches used in each client's local computation. Error bars indicate the 10th and 90th percentiles. All runs are equalized to process the same number of tokens in total.

| Algorithm | Loss | Batches per Client ($\tau$) | | | |
|-----------|------|---|---|---|---|
| | | 1 | 4 | 16 | 64 |
| FedAvg | Pre-Personalization | 3.6 | 3.8 | 4.3 | 5.2 |
| | Post-Personalization | 3.8 | 0.006 | 0.007 | 0.007 |
| FedSGD | Pre-Personalization | 3.6 | 3.7 | 3.9 | 4.2 |
| | Post-Personalization | 3.9 | 3.5 | 3.3 | 3.3 |

Table 11: Median pre-personalization and post-personalization loss after training with FedAvg and FedSGD, with different numbers of batches per client, keeping the total number of tokens processed constant.

# E  Memory Usage

In this section, we provide details of how much memory are used by the various dataset formats in Section 3.1. Recall that we consider three formats: in-memory, hierarchical, and streaming. We compare the amount of time required to iterate over various federated datasets in these formats in Table 3. Here, we instead detail the peak memory usage (in megabytes) when iterating over the same datasets. We note that these were collected on a single CPU, and do not include the time to do operations like shuffling or batching.

The results are in Table 12. We see that in-memory formats use much larger amounts of memory, as they load the entire dataset into memory. By contrast, hierarchical and streaming formats use significantly less peak memory. Moreover, their peak memory usage does not scale with the total size of the dataset, unlike in-memory formats. We note that streaming can use slightly more memory, though only at most 2 MB more in all experiments.

| Dataset Format | In-Memory | Hierarchical | Streaming |
|----------------|-----------|--------------|-----------|
| CIFAR-100 | 156 | 0.40 | 0.74 |
| FedCCnews | 1996 | 0.08 | 1.16 |
| FedBookCO | 6643 | 0.001 | 0.10 |

Table 12: Peak memory usage, in megabytes, when iterating over federated datasets. We iterate over all examples in all group datasets, in serial, on a single CPU. We compare a federated CIFAR-100 dataset (partitioned across 100 groups, each with 100 examples), FedCCnews (in which examples are split across users at a domain level), and FedBookCO (in which examples are split across users at a title level). See Section 4 for more details on the latter two datasets.

