# OpenReview forum: "Towards Federated Foundation Models: Scalable Dataset Pipelines for Group-Structured Learning"
_NeurIPS.cc/2023/Track/Datasets_and_Benchmarks — NeurIPS 2023 Datasets and Benchmarks Poster_

### Official Review · Reviewer_YDTj · 2023-07-06
**Review for Towards Federated Foundation Models: Scalable Dataset Pipelines for Group-Structured Learning**

**Rating:** 8
**Confidence:** 4
**Correctness:** Yes, everything is technically sound …

**Strengths:**

- The authors address an important gap: large-scale federated datasets. Current work uses mostly small-scale datasets, and insights may be limited in their generalization to larger-scale scenarios.
- The authors show the utility of their library by generating four large-scale federated datasets and applying them to evaluate two FL algorithms.


**Additional Feedback:**

Thank you for that interesting work. I would really love to hear your comments on my question about how to make the tool better usable beyond needing manual specifications of split keys. Thinking from the perspective of an FL researcher, I would love to experiment with different ways of splitting the dataset, and I would expect the tool to provide me some advanced support in doing so.


**Clarity:**

The paper is very well written.


**Documentation:**

Documentation is provided in a code repository and in the supplemental material.


**Limitations:**

There is a discussion of limitations w.r.t. the performance of in-memory vs. streaming datasets. I would suggest to also discuss possible limitations w.r.t. the support for automatically creating meaningful distributions of data items into groups. It appears to me that there is no further tooling to assist the users in that process, beyond the possibility to specify the split function.



**Opportunities For Improvement:**

- While it is a useful feature that users can specify their own way of partitioning the input dataset into groups, it would probably also be useful that an automated partitioning, e.g., on statistical properties specified by a user, could be generated. This would help users to systematically study the performance of FL systems under different data partitionings (IID vs. different degrees of “non-IID”). It wasn’t clear to me whether Dataset Grouper provides such a functionality.


**Relation To Prior Work:**

Prior work is clearly discussed.


**Summary And Contributions:**

The paper introduces Dataset Grouper, a library to create group-structured (federated) datasets out of existing datasets based on user-specified partitions. Additionally, the paper introduces four new large-scale federated datasets(FedC4, FedWiki, FedBookCO, FedCCnews) that have been created with Dataset Grouper and uses them to evaluate two well-known FL algorithms: FedAvg and FedSGD. Based on the large-scale datasets used, new insights on FedAvg and FedSGD are generated.

---

> ### Author Response · Authors · 2023-08-22
> **Author Response to Reviewer YDTj**
>
> ​​We thank the reviewer for a positive review and for identifying the advantages and potential impact of Dataset Grouper.
> While we focus mainly on natural non-i.i.d. partitions, Dataset Grouper can support any embarrassingly parallel partitioning functions. To illustrate this generality, we have added two example scripts that can be used to study the effect of the degree of heterogeneity:
> - [group_randomly.py](https://github.com/google-research/dataset_grouper/blob/main/dataset_grouper/examples/datasets/group_randomly.py): assigns each example to a client uniformly at random.
> - [group_by_dirchlet_process.py](https://github.com/google-research/dataset_grouper/blob/main/dataset_grouper/examples/datasets/group_by_dirichlet_process.py): assigns each example to a client based on a Dirichlet process. This is an embarrassingly parallel version of the popular LDA-based method that is popular in the federated learning literature (e.g. [Hsu et al. 2019](https://arxiv.org/abs/1909.06335)).
>
> We note in both cases, the partitioning logic is a simple Python function, generally at most 5-10 lines. These scripts can be used as follows (see their documentation for more details):
>
> ```
> # Randomly partition FedC4 across 1 million clients
>
> python -m group_randomly.py \
>   --data_dir="${DATA_DIR}" \
>   --output_path=/tmp/dataset_grouper/fedc4 \
>   --file_prefix="fedc4_train_random.tfrecord" \
>   --split="train"
>   --num_groups=1000000
>
> # Dirichlet Process partitioning of CIFAR-100 into 100 groups (note that this requires labels or a similar categorical field)
>
> python -m group_by_dirchlet_process.py \
>   --tfds_name=cifar100 \
>   --dirichlet_parameter=0.1 \
>   --group_feature=label \
>   --feature_max=9 \
>   --num_groups=100 \
>   --output_path=/tmp/dataset_grouper/cifar100 \
>   --file_prefix="cifar100_train_dirichlet.tfrecord" \
>   --split="test"
> ```
>
> > **I would love to experiment with different ways of splitting the dataset, and I would expect the tool to provide me some advanced support in doing so.**
>
> We hope the additional examples we have provided above are helpful. If you have further suggestions on additional tooling we can provide, we would be delighted to hear and support them!

---

> > ### Comment · Reviewer_YDTj · 2023-08-26
> >
> > Thanks a lot for providing the additional examples. I think they will be helpful for practitioners to quickly get started with the tool.

---

### Official Review · Reviewer_T2tv · 2023-07-20
**Review comments for Towards Federated Foundation Models**

**Rating:** 8
**Confidence:** 4
**Correctness:** The evaluation and experiment design …
**Clarity:** The paper is clearly written

**Strengths:**

The authors have demonstrated the library's capabilities through large-scale federated language modeling simulations on datasets that are significantly larger than those used in previous work. The paper introduces a novel library that enables the creation of large-scale group-structured datasets/benchmarks in federated settings. Through a succinct yet insightful experiment, the authors demonstrate the library's unique ability to elicit new insights that were not as clearly illustrated by previous benchmarks. This underscores the library's potential to significantly enhance our understanding of federated learning.

**Additional Feedback:**

nil

**Documentation:**

The documentation is sufficient.

**Ethics:**

nil

**Limitations:**

nil

**Opportunities For Improvement:**

The authors of the paper are transparent about the limitations of their work. They acknowledge that the design of the Dataset Grouper involves significant trade-offs. For instance, in many settings, in-memory datasets can be substantially more efficient than streaming datasets. They provide an in-depth discussion of these trade-offs in Section 3 of the paper.

**Relation To Prior Work:**

To the best of my knowledge, I am not aware of any other benchmark for federated foundation models.

**Summary And Contributions:**

This paper proposed a library called Dataset Grouper. The proposed library is designed to create large-scale group-structured datasets for federated learning simulations. It allows for the creation of group-structured versions of existing datasets based on user-specified partitions, providing both flexibility and scalability.

---

> ### Author Response · Authors · 2023-08-22
> **Author Response to Reviewer T2tv**
>
> We thank the reviewer for the positive review and for identifying the potential impact of Dataset Grouper.
>
> To further emphasize the scalability and broad applicability of Dataset Grouper for federated foundation models, we ran two new experiments. First, we scaled our federated training experiments to a 1B parameter model. Second, we evaluate the personalization of a FedC4-trained model on FedBookCO, FedCCNews, and FedWiki, i.e., under various distribution shifts. Both experiments reveal similar trends to those demonstrated in the paper. We summarize these results below and refer to the revised Section 5 for details.
>
> **Experiments on a 1B transformer LM**:
>
> | Algorithm | Pre-personalization loss |
>  |-----------|--------------------------|
> | FedAvg | 4.04 |
> | FedSGD | 3.89 |
>
> **Evaluation on FedBookCO**:
>
> |                      | FedAvg          | FedSGD          |
> |----------------------|-----------------|-----------------|
> | Pre-Personalization  | $5.03 \pm 0.36$ | $4.28 \pm 0.32$ |
> | Post-Personalization | $2.93 \pm 0.74$ | $4.05 \pm 0.33$ |
>
>
> Thank you again! We are happy to answer further questions and comments.

---

> > ### Comment · Reviewer_T2tv · 2023-08-30
> >
> > I will keep my score. Thanks

---

### Official Review · Reviewer_Pkpg · 2023-07-21
**A library for efficient and scalable dataset creation**

**Rating:** 6
**Confidence:** 4
**Correctness:** I believe the methodology used for th…
**Clarity:** It's a clear paper, but I would expec…

**Strengths:**

provide an efficient data processing library to create four large federated datasets to enable large-scale model-dataset pair training.

**Additional Feedback:**

Please refer to the sections above.

**Documentation:**

The dataset description is in great detail in this paper.

**Limitations:**

1. what's the shortcoming of limiting the data access pattern? Will restricting the data access pattern affect the client selection?
2. in table 3, the time of iterating over the whole dataset seems to be incomparable to training time. if it's not the bottleneck, is it really import want to optimize for the data loading?
3. it's unclear about the comparison between different data formats, for other formats, we can also do prefetching when the training is going on for the previous batch, the training time seems to be longer, especially when training a large model.

**Opportunities For Improvement:**

1. the paper is unclear about the difficulty of user-specified partition methods, which might be important for this tool.
2. this paper is unclear about the comparison between different dataset formats, also it's not clear how they do the prefetching and other optimization on the streaming format.




**Relation To Prior Work:**

yes

**Summary And Contributions:**

Dataset Grouper is a library that allows you to create group-structured datasets for federated learning simulation. It works by taking an existing dataset and partitioning it into groups based on user-defined partition functions. The main contribution of this paper,

1. provide a scalable way to partition large federated learning dataset
2. provide four large-scale federated text dataset

---

> ### Author Response · Authors · 2023-08-22
> **Author Response to Reviewer Pkpg: Part 1/2**
>
> We thank the reviewer for a thoughtful review. We address each comment in turn below.
>
> > **Improvement 1: the paper is unclear about the difficulty of user-specified partition methods**
>
> Dataset Grouper can be used with any embarrassingly parallel user-defined partitioning function of the signature `get_key_fn(example) -> group_id`. For instance, the following simple function can be used to partition the dataset by the number of words in an example:
>
> ```
> def get_key_fn_words(example):
>     num_words = len(example[“text”].split())
>     return str(num_words).encode(‘utf-8’)  # a unique ID for each group
> ```
> To illustrate the ease of defining new partition functions, we provide additional example scripts in our GitHub repository (the latter two are recent additions in the rebuttal period):
> - [group_by_feature.py](https://github.com/google-research/dataset_grouper/blob/main/dataset_grouper/examples/datasets/group_by_feature.py) Groups examples according to some discrete feature. For example, it can be used to group examples by some metadata, or pathologically partition by label (so that each client gets a single label).
> - [group_randomly.py](https://github.com/google-research/dataset_grouper/blob/main/dataset_grouper/examples/datasets/group_randomly.py): assigns each example to a client uniformly at random.
> - [group_by_dirchlet_process.py](https://github.com/google-research/dataset_grouper/blob/main/dataset_grouper/examples/datasets/group_by_dirichlet_process.py): assigns each example to a client based on a Dirichlet process. This is an embarrassingly parallel version of the popular LDA-based method that is popular in the federated learning literature (e.g. [Hsu et al. 2019](https://arxiv.org/abs/1909.06335)).
>
> In all three cases, the partition function is a simple Python function that can generally be expressed in just a couple of lines.
>
>
> >  **Improvement 2: unclear about the comparison between different dataset format … prefetching**
>
> Our timing comparisons in Table 3 are obtained from iterating over the datasets with no further pipeline optimization, including prefetching. The reported time in this table is entirely the time spent iterating over all examples in the data pipeline.
>
> The streaming format is orders of magnitude faster than the hierarchical format because of data locality. The I/O cost involved in opening and reading a file (i.e. a shard of the dataset) can be quite slow. As illustrated in Figure 2, the streaming format interleaves a few files and goes through them sequentially, thus amortizing the cost of opening each file. Meanwhile, the random accesses of hierarchical format are expensive: each dataset access potentially requires opening and reading a fresh file and then producing data from a requested client.
>
> > **Limitation 1: shortcoming of limiting the data access pattern … affect the client selection**
>
> The streaming access pattern approximates the following process: randomly shuffle all the clients at the start of an epoch and sequentially iterate through this shuffled order (in practice, the shuffling is approximated by interleaving data from shards in random order and using a shuffle buffer to produce the data). The streaming format does not permit random access, i.e., queries of the form "give me client X’s data, followed by client Z’s data" are not supported.
>
> This tradeoff is somewhat unavoidable as formats that enable arbitrary access patterns generally cannot scale to settings required for LLM training. We note that by way of analogy, modern dataset pipelines for training LLMs centrally generally do not enable arbitrary example access. Instead, they use streams of examples with operations like approximate shuffling.
>
> As we show in Table 3, one pass over FedCCNews takes around 4 minutes in the streaming format while the hierarchical format timed out by not completing a pass in even 2 hours. Similarly, the in-memory format cannot scale to large datasets and goes out of memory.

---

> > ### Author Response · Authors · 2023-08-22
> > **Author Response to Reviewer Pkpg: Part 2/2**
> >
> > > **Limitation 2: time of iterating over the whole dataset seems to be incomparable to training time**
> >
> > The fact that data access with the streaming format does not bottleneck model training is a major advantage of our work. This is not guaranteed with other formats as we explain below.
> >
> > The training time and data iteration time can be compared by merging the numbers from Tables 3 and 4. The data access time is around 10% of the total training time for the streaming format, thanks to our careful design (Table 4). The hierarchical format takes >> 30x the processing time of the streaming format (based on Table 3), meaning that **each data access under the hierarchical format would take >> 3x as long as the training time**.
> >
> > > **Limitation 3: other formats … prefetching when the training on the previous batch**
> >
> > As discussed in the preceding paragraph, the data access time under the hierarchical format takes much longer than it would train the previous batch. Thus, **pre-fetching cannot solve the inefficiency of hierarchical formats**.
> >
> > Pre-fetching does not apply to the in-memory format because everything is already in memory (there is nothing left to pre-fetch). This approach does not scale to large datasets; indeed, FedBookCO and FedC4 are too large to fit in memory.
> >
> > Note that the timing calculations above were performed with FedCCNews --- the hierarchical access time for FedC4 is significantly larger due to more extensive dataset sharding, meaning that a new file will potentially be loaded more frequently.
> >
> > **Conclusion**:
> > Thanks again for the review! We hope that we have clarified your questions on pre-fetching and the acute need for the streaming format. We are happy to go over any further questions or comments.

---

> > > ### Author Response · Authors · 2023-08-28
> > > **Requesting a Rebuttal Response**
> > >
> > > Dear Reviewer Pkpg,
> > >
> > > Thank you for your detailed review! We have carefully read your comments and responded to each one in detail.
> > >
> > > As we near the end of the discussion period, we request you to read our responses and kindly update your review accordingly. If we have addressed your concerns, would you consider upgrading your score?
> > >
> > > We would be delighted to answer further any questions. Thank you again!

---

### Official Review · Reviewer_njkF · 2023-07-21
**A novel framework agnostic structured dataset creator for large scale federated learning simulations**

**Rating:** 7
**Confidence:** 3
**Correctness:** The claims are correct.

**Strengths:**

1. Provides a scalable large-scale dataset iterator for datasets that do not fit into memory
2. The tool is robust and capable of being used in different modalities across different frameworks.
3. The tool will make research into large federated language models much easier.
4. The insight gained from large-scale datasets specifically that Federated learning tends toward meta-learning are quite significant.

**Additional Feedback:**

No additional feedback,

**Clarity:**

The paper is readable. Note on Figure 2(a) for Group 1, will the data be $x_{21}, x_{22}$ or $x_{11}, x_{12}$? The figure is not quite understandable. A bit more detail in the caption would help.

**Documentation:**

Documentation is sufficient.

**Ethics:**

No noticeable ethical concerns.

**Limitations:**

1. It is not immediately clear if the library is compatible with multiprocessing workflows when simulating clients on different nodes and gpus.
2. Memory usage and overhead for the datasets are not reported.

**Opportunities For Improvement:**

A part of the abstract of the paper focuses on the relationship between federated learning and meta-learning. But the result from only one dataset is shown. It would be a much more significant insight if it could be duplicated across all the datasets. Additionally, the result is only mentioned but not explained clearly what is gained from such an insight. It would be a good idea to explain what the implication of this result is.

**Relation To Prior Work:**

Prior work is discussed.

**Summary And Contributions:**

The paper presents Dataset Grouper, a library for creating large-scale group-structured datasets for federated learning simulation. It allows user-defined partitions and can handle large datasets that don't fit into memory. Dataset Grouper is framework-agnostic and flexible in data selection. It enables large-scale federated language modeling simulations, revealing that algorithms like FedAvg work more as meta-learning methods at this scale, useful for personalization and task-specific adaptation.

---

> ### Author Response · Authors · 2023-08-22
> **Author Response to Reviewer njkF**
>
> We thank the reviewer for a thoughtful review. We answer each question below.
>
> > **relationship between federated learning and meta-learning … duplicated across all the datasets**
>
> We repeat the pre- and post-personalization evaluation on FedBookCO, FedCCNews, and FedWiki. Qualitatively similar trends hold, with FedAvg producing a better personalized model and FedSGD producing the better global model. Quantitatively, below we give the average pre- and post-personalization loss for the final model checkpoint, as well as the standard deviation across clients. The full results can be found in Appendix D of the revised manuscript.
>
> | FedBookCO            | FedAvg          | FedSGD          |
> |----------------------|-----------------|-----------------|
> | Pre-Personalization  | $5.03 \pm 0.36$ | $4.28 \pm 0.32$ |
> | Post-Personalization | $2.93 \pm 0.74$ | $4.05 \pm 0.33$ |
>
> | FedCCNews            | FedAvg          | FedSGD          |
> |----------------------|-----------------|-----------------|
> | Pre-Personalization  | $5.24 \pm 0.39$ | $4.32 \pm 0.42$ |
> | Post-Personalization | $1.25 \pm 1.58$ | $3.49 \pm 1.07$ |
>
> >  **...what is gained from such an insight...**
>
> Our results challenge the conventional wisdom that FedAvg uniformly dominates FedSGD in a communication-constrained setting. Instead, we find that FedAvg is better at personalization while FedSGD provides a better global model. Further, we suggest default hyperparameters for federated training of transformer-based LLMs (optimizer, learning rate schedule, etc.).
>
> In summary, our findings provide practitioners with an effective guide to selecting the right algorithm based on their use case and minimizing the total computational cost of training and parameter tuning.
>
> > **...is the library compatible with multiprocessing workflows when simulating clients on different nodes and gpus...**
>
> Yes, Dataset Grouper is fully compatible with data- and model-parallel training pipelines. For our experiments, we use 16 TPUs with multi-processing (see "Hardware configuration" in Sec. 5.1 of page 7).
>
> > **Memory usage and overhead**
>
> We have added a table of peak memory usage for each of the three dataset formats while iterating over the data in Appendix E. We reproduce the table below. Notice that the streaming format only requires O(1 MB) of memory irrespective of the size of the original dataset.
>
> |           | In-Memory | Hierarchical | Streaming |
> |-----------|-----------|--------------|-----------|
> | CIFAR-100 | 156       | 0.40         | 0.74      |
> | FedCCNews | 1996      | 0.08         | 1.16      |
> | FedBookCO | 6643      | 0.001        | 0.10      |
>
> > **Note on Figure 2(a) for Group 1, will the data be $x_{21}, x_{22}$ or $x_{11}, x_{12}$? The figure is not quite understandable. A bit more detail in the caption would help.**
>
> Great catch! The examples in blue should indeed be $x_{11}, x_{12}$. We have updated the figure to correct this.
>
> For the in-memory format, the entire dataset is already loaded into memory up front, which allows for arbitrary group access, but is constrained by the memory size. The hierarchical format requires a file system lookup to retrieve the arbitrary clients but is slowed by having to repeatedly index over a large distributed set of files, in arbitrary order. The streaming format disallows random access but instead gives us access to a stream of clients. This is much faster as it can leverage data locality: sequential access into large files is far more efficient than random access.
>
> **Conclusion**: Thank you again for the review! We hope that we have addressed all your concerns. Please do not hesitate to let us know in case of further questions or concerns.

---

> > ### Comment · Reviewer_njkF · 2023-08-24
> > **Reviewer Response to Authors**
> >
> > Thank you for the detailed response. It is refreshing to know that the trends are similar on FedBookCO, FedCCNews, and FedWiki. I believe my concerns have been sufficiently addressed.

---

### Official Review · Reviewer_wrGw · 2023-07-22
**A solid library, may facilitate the development of FL and LLMs, while with unclear application scenarios, limited innovation, and unsufficient experiments.**

**Rating:** 5
**Confidence:** 4
**Correctness:** Please see details in "*Opportunities…

**Strengths:**

1. The organized four datasets are characterized by large amount of groups or large per-example sequence length, enabling federated LLMs training.
2. This dataloader tradeoffs the memory consumption and access speed.
3. This dataloader provides flexibility in choosing the base dataset and implementing partitions, and is framework-agnostic.

**Additional Feedback:**

Please see details in "*Opportunities For Improvement" above.

**Clarity:**

This paper is easy to follow while with some unclear arguments. Please see details in "*Opportunities For Improvement" above.

**Documentation:**

This work provides sufficient information in terms of data collection, organization, availability and maintenance, together with a URL that is publicly accessible.

**Opportunities For Improvement:**

1. **[Novelty]** Loading data in a streaming manner is a common technique when dealing with large datasets [1], which limites the novelty of this work in terms of both system design and implementation.
2. **[Datasets Curation]** The provided four federated datasets are obtained by partitioning four existing textual datasets, and the efforts lies mainly in partitioning them according to some straightforward and existing manners. The authors mainly highlight the library's support for group partition, but there is little discussion of the rationale, effects and trade-offs for FL/LLM training with different splitting or different groups. The lack of in-depth experiments, analyses, and insights in the provided datasets lowers the contribution of the authors to the dataset.
3. **[Unclear targeted scenarios]** Although the four federated datasets are characterized by their scales, it seems that the authors overly focus on the scale of datasets and ignore the application scenarios. It would be nice to have a discussion on the targeted models and targeted application scenrios of these four datasets, respectively. For example, in FedWiki and FedBookCO, there is only one sequence in each group, what kind of application scenarios would this situation occur in?
4. **[Usability in FL finetuning]** The original four datasets (C4, Wiki, Book, CCnews) are already used in many pre-trained LLMs. If the data library is adopted in FL settings where LLMs are already pretrained and only need to be finetuned (which is also the most common situation in real FL setting), the datasets might not be fully reflected the model training performance since the models had already been exposed to these datasets during pretraining. The experiment also trains a decoder-only model from scratch, which only validates the federated datasets in federated pretraining scenarios.
5. **[Experiments w.r.t. datasets and models]** The paper only provides empirical study on FedC4 and a 110M transformer model (a very small size compared to popular LLMs), but the validity of other datasets and other models in FL are unknown. This severely limits the generalizability and applicability of the proposed library, and raises doubts about its contribution. The authors should conduct more extensive experiments on other mentioned datasets and larger LLMs to demonstrate the effectiveness and robustness of their library.


[1] https://huggingface.co/docs/datasets/stream

**Relation To Prior Work:**

This work provides clear explanation to how it differs from the existing contributions by stating out that previous datasets suffer from 1) limited number of groups, 2) limited sequence length or both of them.

**Summary And Contributions:**

This work aims at a meaningful topic in FL and LLM community that provides sufficient group-structured data for FL and LLMS. It focuses the problem that: 1) existing federated datasets are small-scale in terms of "number of groups", "the quantity of data", or "quantity of data per group"; and 2) staging all data in memory requires enormous memory space and only staging the indices of grouped data files will result in slow access speed.
The main contributions of this work lie in:
1. The authors partitioned four existing datasets to form their federated versions.
2. A library for creating and loading group-structured datasets with the tradeoff of memory consumption and efficiency.
However, there are still several concerns in terms of the novelty, experiments and application scenarios.

---

> ### Author Response · Authors · 2023-08-22
> **Author Response to Reviewer wrGw Part 1/2**
>
> We thank the reviewer for a thorough and insightful review. We answer each question in detail below.
>
> >  **[Novelty] Loading data in a streaming manner is a common**
>
> We agree with the reviewer that the concept of "streams" is not itself novel —  we do not claim novelty over this idea. Rather, our contribution is designing and creating a novel streaming format specifically for federated and group-structured datasets.
>
> To the best of our knowledge, existing FL simulation frameworks do not use streaming formats. Two of the most popular, the LEAF benchmark and TFF, use in-memory and hierarchical formats respectively. Our work is in part inspired by the success of streaming dataset formats (e.g. in TensorFlow Datasets and HuggingFace) for enabling scalability, and we seek to bring that to federated and group-structured datasets. Note that this group-structured functionality is not provided by TFDS or HuggingFace, and does not exist elsewhere as far as we are aware.
>
> Our other contribution is to perform a careful analysis of the trade-offs of scalability, efficiency, and flexibility of various design choices to enable federated data pipelines at scale. Our eventual design decision involved the use of two streams: the first over clients of the dataset, and the second over the examples in each client’s dataset. These design choices were critical in enabling the first-ever 100M - 1B parameter models trained on federated data.
>
> As an added benefit, further advances in streaming data formats pioneered by the much larger and faster-paced LLM community can potentially be directly leveraged in generating their federated counterparts via Dataset Grouper.
>
> > **[Datasets Curation] … discussion of the rationale, effects, and trade-offs for FL/LLM training with different splitting or different groups…**
>
> The goal of our experiments was to demonstrate the type of research questions we can tackle with Dataset Grouper. To this end, we focus mainly on natural non-i.i.d. partitions, which are generally of greater interest to researchers and practitioners than synthetic or random partitions. Natural non-i.i.d. data partitions are also often harder to come by.
>
> Dataset Grouper can support any embarrassingly parallel partitioning functions. To illustrate this generality, in addition to the [feature-based partitioning](https://github.com/google-research/dataset_grouper/blob/main/dataset_grouper/examples/datasets/group_by_feature.py) that we used for multiple datasets, we have added two example scripts for studying the effect of the level of data heterogeneity in random partition schemes:
> - [group_randomly.py](https://github.com/google-research/dataset_grouper/blob/main/dataset_grouper/examples/datasets/group_randomly.py): assigns each example to a client uniformly at random.
> - [group_by_dirchlet_process.py](https://github.com/google-research/dataset_grouper/blob/main/dataset_grouper/examples/datasets/group_by_dirichlet_process.py): assigns each example to a client based on a Dirichlet process. This is an embarrassingly parallel version of the popular LDA-based method that is popular in the federated learning literature (e.g. [Hsu et al. 2019](https://arxiv.org/abs/1909.06335)).
>
> A thorough comparison of the various partitioning schemes supported by Dataset Grouper is beyond the scope of this work, in part because the three schemes above only represent a fraction of the possible partitioning schemes. If time permits, we would be happy to add an additional experiment comparing the natural non-i.i.d. partition of FedC4 to a random partition.

---

> > ### Author Response · Authors · 2023-08-22
> > **Author Response to Reviewer wrGw Part 2/2**
> >
> > > **[Unclear targeted scenarios] discussion on the targeted models and targeted application scenarios … only one sequence in each group in FedWiki and FedBookCO**
> >
> > We are excited by the possible application scenarios enabled by research into federated foundation models but wish to emphasize that there is little previous work in this area and few concrete applications with tangible results that we are aware of. Our goal was instead to enable novel research that is not yet supported by existing benchmarks. Our investigation of the meta-learning properties of FedAvg is one such example. Another example of future work enabled by Data Grouper is research into user-level differential privacy, which in turn is desirable for its better compliance with legal regulations such as the EU GDPR and DMA. It is our hope that the community will find Dataset Grouper useful to generate group-partitioned versions of other large-scale datasets (across multiple modalities) to enable novel use cases.
> >
> > We turn to the four existing datasets now. FedBookCO (each client is a book) and FedWiki (each client is a Wikipedia article) map to applications where each client is an "expert" in a certain topic. In the context of modern LLM pipelines where each sequence is broken up into multiple examples of a fixed length, the fact that each client has a single sequence is much less important than the total number of words. This is especially true for FedBookCO where the 10th percentile data sequence length is 24K words, while the maximum sequence length for LLMs today is O(1K) words.
> >
> > FedC4 is typical of a group-structured pre-training (or 2nd stage pre-training) dataset, which might be practically encountered in practice with documents (corporate, medical, legal, etc.) or emails. FedCCNews is a subset of FedC4 with similar long-tailed characteristics and is suitable for faster experimentation of 2nd stage pre-training or fine-tuning. It can also be used to study the effects of pretraining data contamination.
> >
> > > **[Usability in FL finetuning] finetuning… the models had already been exposed to these datasets during pretraining.**
> >
> > We agree, but emphasize that this is a major challenge for the entire field: the type and amount of public data with permissive licenses is limited. Almost all of this is already used to pre-train LLMs (the details of which are not always publicly available). This problem for the broader field of foundation model research does not yet have a satisfactory solution.
> >
> > Our purpose is to release a library that enables the creation of large-scale federated datasets. Our goal was not to create datasets that have no overlap from datasets used in pretrained LLMs, but rather to create tooling for creating federated datasets with desired properties for research.
> >
> > Additionally, we leverage Dataset Grouper to federate existing public datasets and demonstrate how these already give novel insights. We believe the question of which datasets are useful for FL research is an important question, but not one that can be solved immediately. By giving researchers tools to make dataset creation easier, we hope to make this difficult problem more tractable.
> >
> > We also wish to emphasize that our experiments yield interesting results, despite possible overlap in datasets. For example, we have included experiments that pre-train a model on FedC4 and fine-tune it on a variety of datasets, including FedC4’s validation set, FedBookCO, FedCCNews, and FedWikipedia. In *all* cases, the same behavior occurred: FedAvg did worse in terms of pre-personalization, and markedly better in post-personalization. We note that FedAvg and FedSGD were pre-trained on the same corpus, so any overlap in training and personalization datasets was true for both algorithms.
> >
> > > **[Experiments w.r.t. datasets and models] conduct more extensive experiments on other datasets and larger LLMs**
> >
> > Per the reviewer’s suggestion, we have two new experiments.
> >
> > **Federated training at 1B scale**:
> > We repeated the FedSGD/FedAvg comparison on a 1 billion parameter model. Please see the end of Section 5 in the revised manuscript for details. The average pre-personalization loss (across clients) of the models after training is:
> > - FedAvg: 4.04
> > - FedSGD: 3.89
> >
> > **Experiments on other datasets**: We perform pre- and post-personalization on FedBookCO, FedCCNews, and FedWikipedia. The average pre- and post-personalization losses on FedBookCO for the final model checkpoint, as well as the standard deviation across clients, are below. Please see Appendix D of the revised manuscript for results on other datasets.
> >
> > | FedBookCO            | FedAvg          | FedSGD          |
> > |----------------------|-----------------|-----------------|
> > | Pre-Personalization  | $5.03 \pm 0.36$ | $4.28 \pm 0.32$ |
> > | Post-Personalization | $2.93 \pm 0.74$ | $4.05 \pm 0.33$ |
> >
> > **Conclusion**: Thanks again! We hope that we have addressed all your concerns. Please do not hesitate to reach out in case of further questions.

---

> > > ### Author Response · Authors · 2023-08-28
> > > **Requesting a Rebuttal Response**
> > >
> > > Dear Reviewer wrGw,
> > >
> > > Thank you for your detailed review! We have carefully read your comments and responded in detail, including multiple additional experimental results.
> > >
> > > As we near the end of the discussion period, we request you to read our responses and kindly update your review accordingly. If we have addressed your concerns, would you consider upgrading your score?
> > >
> > > We would be delighted to answer further any questions. Thank you again!

---

### Author Response · Authors · 2023-08-22
**Thank you for the thorough reviews!**

We thank the reviewers for their detailed and thoughtful comments. We are delighted that the reviewers appreciated the scalability, efficiency, and potential impact of our library on federated learning research.

We are excited that the reviewers recognize that we "address an important gap: large-scale federated datasets" with a "scalable" library that is "efficient" and flexible, i.e. it is "capable of being used in different modalities across different frameworks." We are pleased that the reviewers appreciate that Dataset Grouper makes "research into large federated language models much easier" and our experiments "demonstrate the library's unique ability to elicit new insights that were not as clearly illustrated by previous benchmarks."

We have run a number of new experiments based on reviewer comments, including federated training of a 1 billion parameter transformer LM on FedC4, the largest federated-trained model to date, as well as pre- and post-personalization evaluations on multiple downstream datasets. Please see the end of section 5 in the revised paper, as well as Appendix D for details (all the revisions are highlighted in blue), though we reproduce the important findings below. We have also benchmarked the peak memory usage of various dataset formats, see Appendix E. These new experiments demonstrate that Dataset Grouper can scale into the billion-parameter regime and that our findings on FedAvg as a meta-learner are robust to distribution shifts and model scales.

Below, we respond to each reviewer individually. We are happy to answer any further questions or comments.

---

### Author Response · Authors · 2023-08-29
**Requesting reviewer response to the rebuttal**

Dear **Reviewer wrGw** and **Reviewer Pkpg**,

We appreciate your detailed reviews. We have made a serious effort to address your concerns, including experiments on larger models and new datasets.

We would be grateful if you could update your reviews based on the new details and experimental results we provided during the discussion phase. Please let us know if you have any questions or further feedback before the discussion deadline, which is in around 36 hours. If your concerns are sufficiently addressed, would you be open to upgrading the scores you assigned?

Thank you again for your time and feedback.

Best,
The authors

---

### Decision · Program_Chairs · 2023-09-22

**Decision:**

Accept (Poster)

**Comment:**

In this submission, the authors presents their Dataset Grouper, a library that enables flexible choices of dataset and partition, scalable and framework-agnostic simulations for federated language models. The presented evaluations also demonstrate its great potentials. To this end, I recommend accepting this submission.

Hope the authors find the discussion with reviewers useful and make this submission a better one.